# Relational Feature Caching for Accelerating Diffusion Transformers

**Byunggwan Son**[1*]   **Jeimin Jeon**[1*]   **Jeongwoo Choi**[1*]   **Bumsub Ham**[1,2†]

[1]Yonsei University
[2]Korea Institute of Science and Technology (KIST)

{byunggwan.son, jeimin, jeongwoo.choi, bumsub.ham}@yonsei.ac.kr

## Abstract

Feature caching approaches accelerate diffusion transformers (DiTs) by storing the output features of computationally expensive modules at certain timesteps, and exploiting them for subsequent steps to reduce redundant computations. Recent forecasting-based caching approaches employ temporal extrapolation techniques to approximate the output features with cached ones. Although effective, relying exclusively on temporal extrapolation still suffers from significant prediction errors, leading to performance degradation. Through a detailed analysis, we find that 1) these errors stem from the irregular magnitude of changes in the output features, and 2) an input feature of a module is strongly correlated with the corresponding output. Based on this, we propose relational feature caching (RFC), a novel framework that leverages the input-output relationship to enhance the accuracy of the feature prediction. Specifically, we introduce relational feature estimation (RFE) to estimate the magnitude of changes in the output features from the inputs, enabling more accurate feature predictions. We also present relational cache scheduling (RCS), which estimates the prediction errors using the input features and performs full computations only when the errors are expected to be substantial. Extensive experiments across various DiT models demonstrate that RFC consistently outperforms prior approaches significantly. Project page is available at https://cvlab.yonsei.ac.kr/projects/RFC

## 1 Introduction

Diffusion models (Ho et al., 2020; Song & Ermon, 2019) have recently achieved remarkable success in generative tasks, such as text-to-image generation (Rombach et al., 2022; Saharia et al., 2022), and video generation (Blattmann et al., 2023a;b). While early approaches (Ho et al., 2020; Dhariwal & Nichol, 2021) relied on U-Net architectures (Ronneberger et al., 2015), recent works have shifted toward diffusion transformers (DiTs) (Peebles & Xie, 2023), which demonstrate superior performance, particularly when model and data scales increase. However, such gains come with significant computational costs, since DiTs require performing expensive forward passes over numerous denoising timesteps, which hinders their practical application. To address this, feature caching approaches (Ma et al., 2024; Wimbauer et al., 2024; Liu et al., 2025b) have emerged, offering a promising solution for reducing redundant computation in DiTs. These methods are based on the observation that the intermediate features in DiTs are highly similar across adjacent timesteps. Specifically, they perform full computations at certain timesteps, cache the output features of computationally expensive modules (*e.g.*, attention and MLP), and exploit the cached features at subsequent timesteps to reduce redundant computations.

Early caching approaches (Ma et al., 2024; Selvaraju et al., 2024) typically reuse cached features directly without adaptation. While this improves efficiency, the discrepancy between cached and fully computed features accumulates over timesteps, which in turn degrades the generation quality. To alleviate this, recent works have proposed forecasting-based methods, that predict features through

---

*Equal contribution.
†Correspondence author.

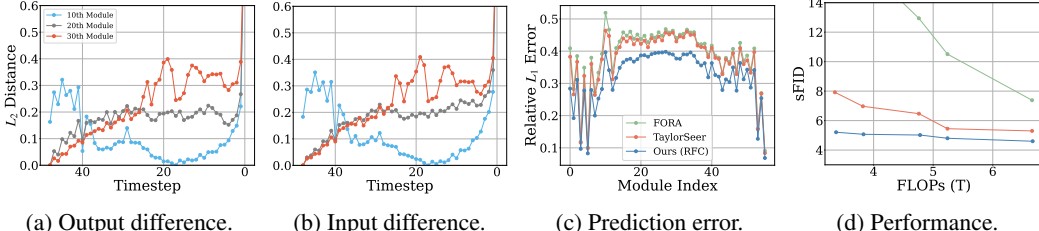

(a) Output difference. (b) Input difference. (c) Prediction error. (d) Performance.

Figure 1: Feature analysis and comparison between existing approaches (FORA (Selvaraju et al., 2024), TaylorSeer (Liu et al., 2025b)), and our method (RFC) using DiT-XL/2 (Peebles & Xie, 2023). (a-b) Min-max normalized $L_2$ distances of output and input features, measured between consecutive timesteps. While the variations of feature changes are irregular, those of input and output remain closely aligned with each other. (c) The prediction errors across different modules. We measure the relative $L_1$ error between output features with and without applying caching methods and average the values over the timesteps. (d) Quantitative results on ImageNet (Deng et al., 2009) evaluated in terms of FLOPs and sFID (Nash et al., 2021).

temporal extrapolation techniques, with an assumption that features evolve smoothly over timesteps. For instance, FasterCache (Lv et al., 2025) and GOC (Qiu et al., 2025) perform a linear extrapolation technique based on features from the two most recent full-compute steps, while TaylorSeer (Liu et al., 2025b) leverages the Taylor expansion to estimate feature changes along timesteps. However, our observations reveal that the magnitude of changes in the output features differs significantly across timesteps (Fig. 1(a)), leading to substantial prediction errors (Fig. 1(c)). Specifically, TaylorSeer achieves slight reductions in prediction error compared to FORA (Selvaraju et al., 2024), which reuses features without adaptation. This degrades the generation quality severely, especially when employing large temporal intervals between full computations (*i.e.*, lower FLOPs in Fig. 1(d)).

To address this limitation, we propose relational feature caching (RFC), a simple yet effective framework that leverages the relationship between the input and output features of modules (*e.g.*, attention and MLP), rather than relying solely on temporal extrapolation techniques. To this end, we propose relational feature estimation (RFE), which estimates the magnitude of changes in the output feature by exploiting the differences in input features. RFE is based on the observation that the magnitude of changes in the output features is highly correlated with that of the input features (Figs. 1(a-b)), enhancing the accuracy of the feature estimation. We also introduce relational cache scheduling (RCS), a dynamic caching strategy that determines when to perform full computations based on estimated output errors. Since directly measuring output errors requires full computations, we instead employ errors in the input prediction as an efficient proxy, leveraging the relationship between the input and output features. Extensive experiments show that our proposed RFC consistently outperforms previous caching methods across a variety of DiT models, demonstrating the effectiveness of our method. We summarize our contributions as follows:

- We propose RFE, a forecasting method that leverages input feature variations to more accurately estimate output features.
- We propose RCS, a dynamic strategy that performs full computations adaptively by estimating the prediction error of the output features from that of input features.
- Extensive experiments across various DiT models demonstrate that using RFE and RCS consistently outperforms existing caching methods in terms of both the generation quality and computational efficiency.

## 2 RELATED WORK

In this section, we review representative works related to accelerating diffusion models.

### 2.1 TIMESTEP REDUCTION

The iterative generation process of diffusion models (Ho et al., 2020; Song & Ermon, 2019) leads to a slow inference and high computational costs, as each sample requires hundreds or even thousands of denoising steps. To mitigate this, many works attempt to reduce the number of timesteps.

DDIM (Song et al., 2020) introduces a non-Markovian deterministic sampler that preserves the DDPM (Ho et al., 2020) training objective, while skipping intermediate timesteps. DPM-Solver (Lu et al., 2022) extends this idea with higher-order ODE solvers, approximating nonlinear terms through the Taylor expansion. While these approaches achieve a faster sampling, their performance degrades significantly when the number of steps becomes extremely small.

Another group of methods distill a pretrained diffusion model into a student model that generates comparable results with only a few denoising steps. Progressive distillation (Salimans & Ho, 2022) iteratively trains a student model, which then becomes the teacher for the next stage with fewer steps. Consistency models (Song et al., 2023) learn direct mappings from noisy images at arbitrary timesteps to noise-free images, enabling skipping intermediate denoising steps. Although these approaches generate high-quality images with limited timesteps, they typically incur a significant training overhead.

## 2.2 FEATURE CACHING

Feature caching methods aim to reduce the computational cost across timesteps by storing fully computed features at certain timesteps and exploiting them for subsequent steps to avoid redundant computations. Early approaches (Ma et al., 2024; Selvaraju et al., 2024; So et al., 2024) use uniform intervals for full computations, but this ignores the fact that cache errors can vary significantly across timesteps. To address this, CacheMe (Wimbauer et al., 2024) sets a schedule for the full computation before the denoising process based on the cache error, but this predetermined schedule does not consider varying cache errors across samples. To mitigate this, TeaCache (Liu et al., 2025a) caches input features at full-compute steps and adaptively triggers full computations when the current input feature deviates significantly from the last cached feature. It however requires an additional calibration step to predict output cache errors from input differences. Furthermore, all the aforementioned caching approaches reuse the cached features directly without an adaptation, which can cause substantial cache errors, particularly under long intervals between successive full computations. The works of (Zou et al., 2025; 2024) aim to reduce cache errors of important tokens by selectively performing full computations on them, while still relying on cached features for the majority of other tokens.

More recent works address the cache error by predicting features from cached ones rather than reusing them directly. FasterCache (Lv et al., 2025) observes that the change in the output features is directionally consistent, and uses a linear extrapolation technique to predict subsequent features. GOC (Qiu et al., 2025) also adopts a linear extrapolation technique, while reusing the cached features for timesteps where the extrapolation is less accurate. TaylorSeer (Liu et al., 2025b) leverages the higher-order Taylor expansions to better capture nonlinear feature dynamics, providing more accurate predictions than the linear extrapolation and improving the generation quality.

## 3 METHOD

In this section, we describe diffusion models and feature caching in DiTs (Sec. 3.1). We then introduce our RFC framework in detail (Sec. 3.2).

## 3.1 PRELIMINARIES

**Diffusion models.** Diffusion models consist of a forward and a reverse process. In the forward process, Gaussian noise $\epsilon$, sampled from a normal distribution $\mathcal{N}(0,1)$, is added to the input image $x_0$ over a sequence of timesteps. Specifically, at a given timestep $t$, the noisy image $x_t$ is defined as:

$$x_t = \sqrt{\alpha_t}x_0 + \sqrt{1-\alpha_t}\epsilon, \quad \epsilon \sim \mathcal{N}(0, I), \tag{1}$$

where $\alpha_t$ denotes a predefined noise schedule at timestep $t$. The reverse process recovers the original image $x_0$ from the noisy input by iteratively denoising it through a series of steps. At each timestep $t$, the model predicts the noise component, which is then used to compute the denoised image for the previous timestep $x_{t-1}$. For instance, DDIM sampler (Song et al., 2020) defines the reverse process as follows:

$$x_{t-1} = \sqrt{\alpha_{t-1}} \left( \frac{x_t - \sqrt{1-\alpha_t}\epsilon_\theta(x_t, t)}{\sqrt{\alpha_t}} \right) + \sqrt{1-\alpha_{t-1}}\epsilon_\theta(x_t, t), \tag{2}$$

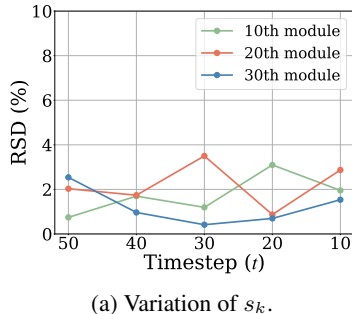 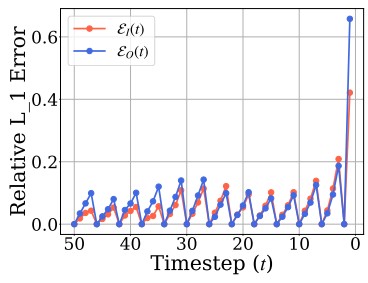

(a) Variation of $s_k$.      (b) Relation between $\mathcal{E}_O$ and $\mathcal{E}_I$.

Figure 2: Empirical analyses in DiT-XL/2 (Peebles & Xie, 2023). (a) RSD of $s_k(t-k)$ with varying $t$. (b) Relative $L_1$ errors of the output and input features, $i.e.$, $\mathcal{E}_O(t)$ and $\mathcal{E}_I(t)$, respectively. Please see the text for details.

where $\epsilon_\theta$ is the noise prediction model parameterized by $\theta$. While early diffusion models (Ho et al., 2020; Dhariwal & Nichol, 2021) adopt U-Net architectures (Ronneberger et al., 2015), DiTs (Peebles & Xie, 2023) instead parameterize $\epsilon_\theta$ with transformers. DiTs have been shown to be effective for large-scale generation tasks, but at the cost of a substantial computational overhead.

**TaylorSeer (Liu et al., 2025b).** TaylorSeer reduces cache errors in diffusion transformers by predicting output features through the Taylor expansion around the last fully computed timestep $t$. Given a cache interval $N$, the $m$-th order prediction of the output feature for the $l$-th module at timestep $t-k$ ($0 < k < N$) is computed as follows:

$$O^l_{\text{Taylor}}(t-k) = O^l(t) + \sum_{i=1}^{m} \frac{k^i}{i!} \frac{\Delta_N^i O^l(t)}{N^i},$$ (3)

where $O^l(t)$ is the output feature of the $l$-th module at timestep $t$, and $\Delta_N^i$ denotes the $i$-th order finite difference operator with the stride $N$ defined recursively as:

$$\Delta_N^i O^l(t) = \begin{cases} O^l(t) - O^l(t+N), & i = 1, \\ \Delta_N^{i-1} O^l(t) - \Delta_N^{i-1} O^l(t+N), & i > 1. \end{cases}$$ (4)

For the clarity and convenience of notation, we omit the superscript $i$ for first-order differences and the module index $l$ in the remainder of this paper.

### 3.2 Relational Feature Caching

While TaylorSeer (Liu et al., 2025b) achieves the state-of-the-art performance over previous caching approaches, it still shows considerable prediction errors (Fig. 1(c)). This is because the magnitude of the feature changes is irregular across the timesteps (Fig. 1(a)), making it difficult to accurately estimate the subsequent features based only on temporal extrapolation techniques. To this end, we propose RFC, a framework that enhances feature prediction by leveraging the relationship between input and output features through two complementary components, RFE and RCS. RFE estimates the magnitude of changes in the output features, while RCS identifies when predictions become unreliable and performs full computation accordingly.

**Relational feature estimation (RFE).** We have observed in Fig. 1(a-b) that the magnitudes of changes in the input and output features for the same module are highly correlated, suggesting that differences in input features can serve as effective predictors of the output variations. To quantify this relationship, we define the ratio between the magnitudes of changes in output and input features as follows:

$$s_k(t-k) = \frac{\|\Delta_k O(t-k)\|_2}{\|\Delta_k I(t-k)\|_2},$$ (5)

where $O(t-k)$ and $I(t-k)$ denote the output and input features of the same module, respectively, at timestep $t-k$, and $\Delta_k$ is the first-order difference with stride $k$ in Eq. (4). To evaluate the consistency of the ratio, we compute $s_k(t-k)$ for $k \in [1, 9]$, and measure the relative standard deviation (RSD)

over the resulting values. We can see from Fig. 2(a) that the RSD values remain consistently low (typically around 2%), indicating that $s_k(t-k)$ is highly consistent across timesteps. To support the empirical consistency of $s_k(t-k)$, we present the following proposition:

**Proposition 1.** *Assume that the mapping from input to output features is locally linear, and the direction of the difference vector $\Delta_k I(t-k)$ remains constant for $1 \leq k \leq N$, where $N$ is an interval between full computations. Then, the ratio $s_k(t-k)$ is approximately invariant w.r.t. $k$.*

*Proof.* Suppose the output feature $O(t)$ is a locally linear to the input feature $I(t)$, *i.e.*,

$$O(t) = AI(t) + b, \tag{6}$$

where $A$ and $b$ denote the weight matrix and bias vector of the linear transformation, respectively. Then, the change in output features over the temporal offset $k$ can be written as:

$$\Delta_k O(t-k) = O(t-k) - O(t) \tag{7}$$
$$= A(I(t-k) - I(t)) \tag{8}$$
$$= A\Delta_k I(t-k). \tag{9}$$

Taking the $L_2$-norm of both sides gives:

$$\|\Delta_k O(t-k)\|_2 = \|A\Delta_k I(t-k)\|_2 \tag{10}$$
$$= \|Au_k(t-k)\|_2 \|\Delta_k I(t-k)\|_2, \tag{11}$$

where $u_k(t-k)$ is the normalized direction of the input change defined as:

$$u_k(t-k) = \frac{\Delta_k I(t-k)}{\|\Delta_k I(t-k)\|_2}. \tag{12}$$

By dividing both sides of Eq. (11) with $\|\Delta_k I(t-k)\|_2$, we have the following:

$$s_k(t-k) = \frac{\|\Delta_k O(t-k)\|_2}{\|\Delta_k I(t-k)\|_2} = \|Au_k(t-k)\|_2. \tag{13}$$

By the assumption that $u_k(t-k)$ remains constant for $1 \leq k \leq N$, the ratio $s_k(t-k)$ is invariant w.r.t. $k$. $\qquad \square$

Note that the two assumptions in Proposition 1 are commonly observed in diffusion models. First, changes in input features are usually small (Ma et al., 2024; Selvaraju et al., 2024; So et al., 2024), which allows for the output features to be locally approximated using Taylor's theorem. Second, the works of (Lv et al., 2025; Qiu et al., 2025) show that the direction of feature changes remains largely consistent across timesteps. We provide empirical validation of the assumptions in Sec. A.

Based on the above finding, we propose RFE, a simple yet effective method that estimates the magnitude of changes in output features $\|\Delta_k O(t-k)\|_2$ by leveraging the changes in input features $\|\Delta_k I(t-k)\|_2$. Specifically, we can rewrite Eq. (5) as follows:

$$\|\Delta_k O(t-k)\|_2 = s_k(t-k)\|\Delta_k I(t-k)\|_2. \tag{14}$$

Given the consistency of $s_k(t-k)$ across timesteps, we can approximate Eq. (14) as follows:

$$\|\Delta_k O(t-k)\|_2 \approx s_N(t)\|\Delta_k I(t-k)\|_2, \tag{15}$$

where $s_N(t)$ is the ratio computed between the two most recent full computations with the interval of $N$. Note that obtaining an input feature $I(t-k)$ is efficient, since it only requires lightweight operations (*e.g.*, LayerNorm (Ba et al., 2016), scaling, and shifting). Consequently, RFE refines the predicted magnitude of feature changes in Eq. (3) as follows:

$$O_{\text{RFE}}(t-k) = O(t) + (s_N(t)\|\Delta_k I(t-k)\|_2)g\left(\sum_{i=1}^{m} \frac{k^i}{i!} \frac{\Delta_N^i O(t)}{N^i}\right), \tag{16}$$

where $g(\cdot)$ is an $L_2$ normalization function as follows:

$$g(x) = \frac{x}{\|x\|_2}. \tag{17}$$

RFE can capture irregular dynamics of feature changes, reducing the error of feature prediction, efficiently and effectively.

**Relational cache scheduling (RCS).** Although RFE significantly improves the accuracy of the feature prediction, estimation errors are unavoidable and tend to fluctuate over timesteps, making a fixed caching interval $N$ suboptimal. To address this, we propose RCS, a dynamic scheduling strategy that determines when to perform a full computation based on the estimated prediction error. To this end, we define the error of the output prediction at timestep $t - k$ as follows:

$$E_O(t - k) = O(t - k) - O_{\text{RFE}}(t - k). \tag{18}$$

However, computing $E_O(t - k)$ is not feasible during sampling, since it depends on $O(t - k)$, which requires a costly computation (*e.g.*, attention or MLP). Instead, we propose to estimate this error using the error of the input prediction $E_I(t - k)$ as a proxy:

$$E_I(t - k) = I(t - k) - I_{\text{Taylor}}(t - k), \tag{19}$$

where the predicted input feature $I_{\text{Taylor}}(t - k)$ is computed with the Taylor expansion as follows[1]:

$$I_{\text{Taylor}}(t - k) = I(t) + \sum_{i=1}^{m} \frac{k^i}{i!} \frac{\Delta_N^i I(t)}{N^i}. \tag{20}$$

Our intuition is that prediction errors in the output tend to increase when the feature changes abruptly, and such changes are highly correlated with the variations in the corresponding input (Figs. 1(a-b) and 2(a)). Therefore, errors in the output prediction are likely to be correlated with those in the input prediction. To support this, we compare the relative $L_1$ error of the output and input features, which is defined as follows:

$$\mathcal{E}_O(t - k) = \frac{\|E_O(t - k)\|_1}{\|O(t - k)\|_1}, \quad \mathcal{E}_I(t - k) = \frac{\|E_I(t - k)\|_1}{\|I(t - k)\|_1}. \tag{21}$$

As shown in Fig. 2(b), the trends of the input and output errors closely align across timesteps. Based on this, we aim to estimate the accumulated errors in the output prediction during the sampling process by tracking the relative $L_1$ error of the input prediction for the first module. We then perform a full computation when the accumulated error exceeds a predefined threshold $\tau$ as follows:

$$\sum_{j=1}^{k} \mathcal{E}_I(t - j) > \tau. \tag{22}$$

The left term in Eq. (22) effectively reflects the accumulation of errors in the output prediction. By adjusting $\tau$, RCS controls the trade-off between the generation quality and efficiency. RCS performs full computations more frequently at timesteps, where the accumulated errors are large, reducing the cache error and improving the generation quality.

## 4 EXPERIMENTS

In this section, we first describe our implementation details (Sec. 4.1). We then provide quantitative and qualitative comparisons of RFC with state-of-the-art methods (Sec. 4.2), and present an in-depth analysis of RFC (Sec. 4.3). More experimental results and a discussion on the limitation of our work can be found in Appendix.

### 4.1 IMPLEMENTATION DETAILS

**Datasets and models.** We apply RFC to various DiT models and perform extensive experiments on standard benchmarks for the class-conditional, text-to-image, and text-to-video generation tasks. For class-conditional generation, we perform experiments using the DiT-XL/2 (Peebles & Xie, 2023) model trained on ImageNet (Deng et al., 2009). We use FLUX.1 dev (Batifol et al., 2025) for text-to-image generation and HunyuanVideo (Kong et al., 2024) for text-to-video generation, and evaluate them on DrawBench (Saharia et al., 2022) and VBench (Huang et al., 2024), respectively. We use the DDIM sampler (Song et al., 2020) with 50 steps for DiT-XL/2, while using the Rectified Flow sampler (Liu et al., 2023) for FLUX.1 dev and HunyuanVideo with the same number of steps.

---

[1]RFE is not applied to input prediction, as it is defined to exploit input–output relationships.

Table 1: Quantitative comparisons on class-conditional image generation for DiT-XL/2 (Peebles & Xie, 2023) on ImageNet (Deng et al., 2009). Numbers in bold indicate the best performance, while underscored ones are the second best among similar computational costs. $N$ is the interval between full computations, and $m$ refers to the order of the Taylor expansion.

| Methods | NFC | FLOPs (T)↓ | Latency (s)↓ | FID↓ | sFID↓ | FID2FC↓ | sFID2FC↓ | IS↑ |
|---|---|---|---|---|---|---|---|---|
| Full-Compute | 50 | 23.74 | 7.61 | 2.32 | 4.32 | - | - | 241.25 |
| FORA ($N = 4$) (Selvaraju et al., 2024) | 14 | 6.66 | 2.37 | 4.33 | 7.38 | 1.90 | 6.32 | 215.33 |
| ToCa ($N = 4$) (Zou et al., 2025) | 17 | 8.73 | 3.64 | 3.03 | 5.02 | 0.86 | 3.56 | 229.01 |
| DuCa ($N = 4$) (Zou et al., 2024) | 17 | 7.66 | 2.94 | 3.39 | 4.95 | 1.19 | 4.22 | 224.03 |
| TaylorSeer ($N = 4, m = 2$) (Liu et al., 2025b) | 14 | 6.66 | 2.84 | 2.55 | 5.30 | 0.44 | 2.17 | **232.26** |
| RFC ($m = 2$) | 14.01 | 6.67 | 2.86 | **2.52** | **4.60** | **0.30** | **1.33** | 231.00 |
| FORA ($N = 5$) (Selvaraju et al., 2024) | 11 | 5.24 | 2.03 | 6.12 | 10.51 | 3.57 | 10.19 | 196.04 |
| ToCa ($N = 5$) (Zou et al., 2025) | 14 | 7.43 | 3.35 | 3.53 | 5.36 | 1.28 | 4.50 | 224.07 |
| DuCa ($N = 5$) (Zou et al., 2024) | 14 | 6.32 | 2.57 | 6.02 | 6.71 | 4.19 | 8.18 | 196.96 |
| TaylorSeer ($N = 5, m = 2$) (Liu et al., 2025b) | 11 | 5.24 | 2.52 | **2.70** | 5.45 | 0.56 | 2.74 | **228.76** |
| RFC ($m = 2$) | 11.01 | 5.24 | 2.52 | 2.71 | **4.80** | **0.48** | **2.14** | 227.72 |
| FORA ($N = 6$ (Selvaraju et al., 2024)) | 10 | 4.76 | 1.82 | 8.12 | 12.94 | 5.32 | 12.87 | 179.11 |
| ToCa ($N = 6$) (Zou et al., 2025) | 13 | 7.01 | 3.27 | 3.54 | 5.54 | 1.44 | 5.18 | 226.32 |
| DuCa ($N = 6$) (Zou et al., 2024) | 13 | 5.86 | 2.53 | 6.37 | 6.74 | 4.51 | 7.86 | 187.74 |
| TaylorSeer ($N = 6, m = 2$) (Liu et al., 2025b) | 10 | 4.76 | 2.27 | 3.10 | 6.47 | 1.00 | 4.63 | 222.85 |
| RFC ($m = 2$) | 10.04 | 4.78 | 2.46 | **2.75** | **5.02** | **0.54** | **2.43** | **226.93** |
| FORA ($N = 7$) (Selvaraju et al., 2024) | 8 | 3.35 | 1.73 | 12.63 | 18.49 | 9.66 | 19.11 | 147.06 |
| ToCa ($N = 7$) (Zou et al., 2025) | 14 | 6.25 | 2.90 | 4.04 | 5.72 | 1.62 | 4.72 | 214.25 |
| DuCa ($N = 7$) (Zou et al., 2024) | 14 | 4.96 | 2.24 | 7.76 | 7.77 | 5.45 | 8.80 | 181.50 |
| TaylorSeer ($N = 7, m = 2$) (Liu et al., 2025b) | 8 | 3.82 | 2.02 | 3.46 | 6.97 | 1.30 | 5.61 | 217.18 |
| RFC ($m = 2$) | 8.02 | 3.83 | 2.15 | **3.12** | **5.07** | **0.81** | **3.10** | **219.20** |
| TaylorSeer ($N = 9, m = 2$) (Liu et al., 2025b) | 7 | 3.35 | 1.89 | 4.90 | 7.92 | 2.33 | 7.35 | 198.46 |
| RFC ($m = 2$) | 7.04 | 3.37 | 1.99 | **3.40** | **5.21** | **1.03** | **3.66** | **215.39** |

**Evaluation and metric.** Following the previous approaches (Zou et al., 2025; Liu et al., 2025b), we evaluate 50K images of size 256×256 for class-conditional generation using standard metrics, including Fréchet Inception Distance (FID) (Heusel et al., 2017), sFID (Nash et al., 2021), and Inception Score (IS) (Salimans et al., 2016). In addition, we introduce FID2FC and sFID2FC, which quantify the degradation introduced by caching through measuring the FID and sFID scores, respectively, between the images obtained from full computations and feature caching methods. For the text-to-image generation task, we generate 200 images of size 1000×1000 from DrawBench prompts (Saharia et al., 2022) and evaluate the image quality and text-to-image alignment using ImageReward (Xu et al., 2023) and the CLIP score (Hessel et al., 2021), respectively. We also evaluate PSNR, SSIM (Wang et al., 2004), and LPIPS (Zhang et al., 2018) computed between images generated by full computations and caching approaches. For text-to-video generation, we produce 2,838 videos by generating three videos for each of the 946 prompts. We evaluate the generated videos with the VBench score (Huang et al., 2024), as well as PSNR, SSIM, and LPIPS. For all tasks, we report FLOPs and latency to evaluate the computational efficiency. For a fair comparison, we reproduce the results of state-of-the-art methods using the official source codes, and adjust the threshold $\tau$ in Eq. (22) to ensure that the average number of full computations (NFC) matches that of other methods.

## 4.2 RESULTS

**Quantitative results.** We show in Tables 1-3 quantitative comparisons of RFC and state-of-the-art methods (Selvaraju et al., 2024; Zou et al., 2025; 2024; Liu et al., 2025b) for class-conditional image, text-to-image, and text-to-video generation tasks, respectively. We summarize our findings as follows: (1) RFC outperforms existing caching approaches across various generative tasks, by a large margin. For example, we can see from Table 1 that RFC with 3.37 TFLOPs outperforms TaylorSeer (Liu et al., 2025b) with 4.76 TFLOPs by 1.26 in terms of sFID. We can also see that the improvement is significant for metrics that compare the quality of generated images from full computations and caching approaches (*e.g.*, FID2FC, sFID2FC, PSNR, SSIM (Wang et al., 2004), and LPIPS (Zhang et al., 2018)), suggesting that RFC can accurately estimate output features by leveraging the relationship with the corresponding input features. (2) RFC performs particularly

Table 2: Quantitative comparisons on text-to-image generation for FLUX.1 dev (Batifol et al., 2025) on DrawBench (Saharia et al., 2022). Numbers in bold indicate the best performance, while underscored ones are the second best among similar computational costs. $N$ is the interval between full computations, and $m$ refers to the order of the Taylor expansion.

| Methods | NFC | FLOPs (T)↓ | Latency (s)↓ | PSNR↑ | SSIM↑ | LPIPS↓ | IR↑ | CLIP↑ |
|---|---|---|---|---|---|---|---|---|
| Full-Compute | 50 | 2813.50 | 26.210 | - | - | - | 0.9655 | 17.0720 |
| FORA ($N = 3$) (Selvaraju et al., 2024) | 17 | 957.33 | 10.489 | 17.4967 | 0.7236 | 0.4209 | 0.9111 | 16.9515 |
| ToCa ($N = 5$) (Zou et al., 2025) | 14 | 1056.26 | 18.591 | 18.0333 | 0.7269 | 0.3490 | 0.9457 | 17.0063 |
| TaylorSeer ($N = 4, m = 1$) (Liu et al., 2025b) | 14 | 788.59 | 8.863 | 19.5722 | 0.7695 | 0.3226 | 0.9204 | 17.0659 |
| TaylorSeer ($N = 4, m = 2$) | 14 | 788.59 | 8.985 | 19.7733 | 0.7706 | 0.3175 | 0.9407 | 17.0527 |
| RFC ($m = 1$) | 14.02 | 789.82 | 9.148 | 20.2707 | 0.7905 | 0.2934 | 0.9448 | 17.0288 |
| RFC ($m = 2$) | 13.80 | 777.44 | 9.364 | 20.3524 | 0.7930 | 0.2952 | 0.9499 | 17.0394 |
| FORA ($N = 7$) (Selvaraju et al., 2024) | 8 | 451.10 | 6.065 | 15.8656 | 0.6692 | 0.5557 | 0.6121 | 16.8531 |
| ToCa ($N = 12$) (Zou et al., 2025) | 5 | 673.88 | 11.504 | 16.2240 | 0.6479 | 0.5512 | 0.6238 | 16.9020 |
| TaylorSeer ($N = 9, m = 1$) (Liu et al., 2025b) | 8 | 451.10 | 6.382 | 16.1346 | 0.6736 | 0.5261 | 0.7602 | 17.0769 |
| TaylorSeer ($N = 9, m = 2$) | 8 | 451.10 | 6.796 | 16.5492 | 0.6563 | 0.5328 | 0.7996 | 17.0534 |
| RFC ($m = 1$) | 8.00 | 451.23 | 6.407 | 16.7952 | 0.6979 | 0.4645 | 0.9142 | 17.0165 |
| RFC ($m = 2$) | 8.03 | 452.91 | 6.849 | 16.9188 | 0.6936 | 0.4708 | 0.9189 | 16.9641 |

Table 3: Quantitative comparisons on text-to-video generation for HunyuanVideo (Kong et al., 2024) on VBench (Huang et al., 2024). Numbers in bold indicate the best performance among similar computational costs. $N$ is the interval between full computations, and $m$ refers to the order of the Taylor expansion.

| Methods | NFC | FLOPs (T)↓ | Latency (s)↓ | PSNR↑ | SSIM↑ | LPIPS↓ | VBench↑ |
|---|---|---|---|---|---|---|---|
| Full-Compute | 50 | 7520.00 | 263.480 | - | - | - | 81.40 |
| FORA ($N = 6$) (Selvaraju et al., 2024) | 9 | 1359.19 | 55.339 | 15.5221 | 0.4349 | 0.2743 | 78.51 |
| TaylorSeer ($N = 5, m = 1$) (Liu et al., 2025b) | 10 | 1510.21 | 60.051 | 16.3606 | 0.5126 | 0.1985 | 80.77 |
| TaylorSeer ($N = 6, m = 1$) | 9 | 1359.19 | 55.872 | 15.5321 | 0.4606 | 0.2448 | 79.52 |
| RFC ($m = 1$) | 8.96 | 1354.65 | 57.786 | 18.5408 | 0.6352 | 0.1329 | 80.83 |
| FORA ($N = 8$) (Selvaraju et al., 2024) | 7 | 1058.45 | 48.340 | 14.8878 | 0.4114 | 0.2959 | 77.50 |
| TaylorSeer (Liu et al., 2025b) ($N = 8, m = 1$) | 7 | 1058.45 | 48.736 | 15.1999 | 0.4408 | 0.2624 | 79.59 |
| RFC ($m = 1$) | 7.09 | 1072.65 | 49.684 | 18.2521 | 0.6155 | 0.1436 | 80.49 |

well under limited computational budgets. In such cases, prior approaches suffer from large errors in feature prediction, mainly due to irregular changes of features across timesteps, which degrades the performance. Even with limited computation, RFC accurately estimates the output features with RFE and effectively performs full computations with RCS, preserving the generation quality. (3) With a similar NFC, the overhead of RFC compared to TaylorSeer (Liu et al., 2025b) is minimal. This is because computing the input features in RFC only requires lightweight operations such as LayerNorm (Ba et al., 2016), shifting, and scaling, which adds negligible computational costs, while improving the generation quality significantly.

**Qualitative results.** We show in Fig. 3 the qualitative comparisons of images generated using DiT-XL/2 (Peebles & Xie, 2023) and FLUX.1 dev (Batifol et al., 2025). We observe that RFC produces more realistic images that are visually closer to those generated by full computations. For example, in the first row of Fig. 3(right), RFC better preserves the brick structure compared to the prior approaches (Selvaraju et al., 2024; Liu et al., 2025b), closely matching the image generated by full computations, indicating that RFC effectively minimizes the prediction errors in the output features. We also show in Fig. 4 generated videos using HunyuanVideo (Kong et al., 2024). We can see that RFC produces high-quality videos that are similar to those generated by full computations, demonstrating the effectiveness of RFC once more. Please see Appendix for more results.

## 4.3 DISCUSSION

**Ablations.** We show in Table 4 the ablation study for each component of RFC. We can see that applying either RFE or RCS alone already outperforms TaylorSeer (Liu et al., 2025b) regardless of NFC. For instance, when NFC is 11, applying RFE and RCS to TaylorSeer reduces the sFID

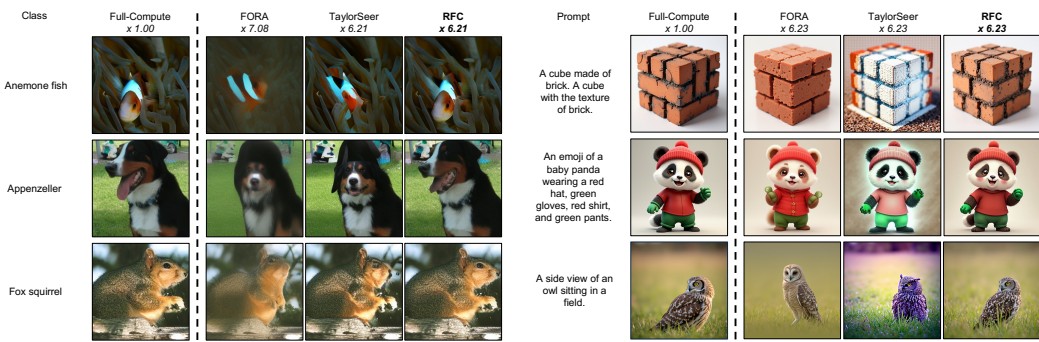

Figure 3: Qualitative comparisons of (left) class-conditional image generation for DiT-XL/2 (Peebles & Xie, 2023) on ImageNet (Deng et al., 2009), and (right) text-to-image generation for FLUX.1 dev (Batifol et al., 2025) on DrawBench (Saharia et al., 2022).

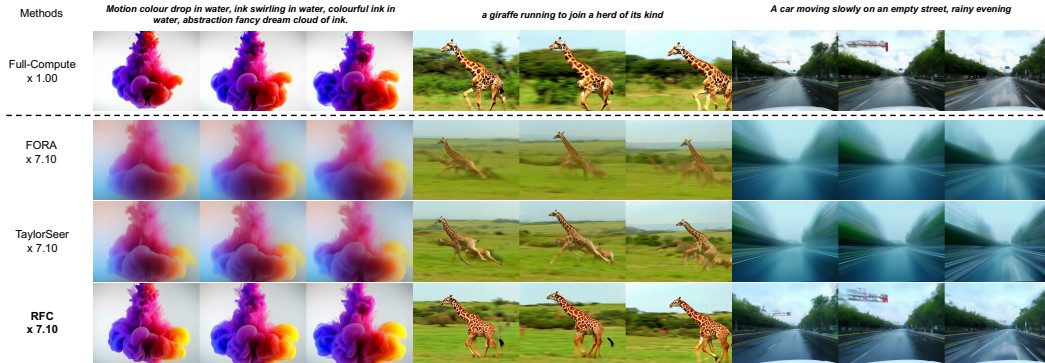

Figure 4: Qualitative comparisons of text-to-video generation for HunyuanVideo (Kong et al., 2024) on VBench (Huang et al., 2024). Please see the supplementary material for the actual video clips.

from 5.85 to 5.22 and 5.21, respectively. This indicates that both RFE and RCS can effectively improve feature prediction by leveraging the input–output feature relationship. Moreover, combining both components (*i.e.*, RFC) further reduces the sFID to 4.88, suggesting that RFE and RCS are complementary to each other.

**Relational feature estimation (RFE).** Feature forecasting methods based on linear extrapolation techniques (Lv et al., 2025; Qiu et al., 2025) can be formulated as:

$$O_{\text{pred}}(t - k) = O(t) + \frac{k}{N} w(t) \Delta_N O(t) \tag{23}$$

where $O_{\text{pred}}(t-k)$ is the predicted output feature at timestep $t-k$, $\Delta_N O(t)$ is the difference between features from the two latest full-compute steps, $N$ is the interval between the full computations, and $w(t)$ is a scaling factor at the timestep $t$. Existing approaches differ in how $w(t)$ is determined. For instance, FasterCache (Lv et al., 2025) linearly increases $w(t)$ from zero to one as $t$ increases, while GOC (Qiu et al., 2025) employs a fixed hyperparameter for $w(t)$. We show in Table 5 quantitative comparisons between RFE and these strategies. We can see that RFE consistently outperforms all the others significantly. This suggests that a simple linear extrapolation fails to capture the irregular changes in the output features, while RFE improves the prediction by leveraging the input–output relationship.

**Relational cache scheduling (RCS)** RCS performs full computations based on the prediction error in input features of the first module. To evaluate the effectiveness of RCS, we provide in Table 6 quantitative results of various scheduling strategies on ImageNet (Deng et al., 2009). First, we adopt a distance-based strategy (Liu et al., 2025a) and schedule full computations based on the distance

Table 4: Quantitative comparison of different components for DiT-XL/2 (Peebles & Xie, 2023) on ImageNet (Deng et al., 2009). All results are obtained using the first-order approximation ($m = 1$).

| Methods | NFC | FID↓ | sFID↓ | FID2FC↓ | sFID2FC↓ |
|---|---|---|---|---|---|
| TaylorSeer (Liu et al., 2025b) | 14 | 2.65 | 5.60 | 0.57 | 2.77 |
| RFE | 14 | 2.52 | 5.18 | 0.43 | 2.02 |
| RCS | 14 | 2.52 | 4.76 | 0.36 | 1.88 |
| RFC (RFE + RCS) | 14 | 2.51 | 4.66 | 0.31 | 1.41 |
| TaylorSeer (Liu et al., 2025b) | 11 | 2.87 | 5.85 | 0.73 | 3.53 |
| RFE | 11 | 2.69 | 5.22 | 0.55 | 2.55 |
| RCS | 11 | 2.77 | 5.21 | 0.62 | 3.09 |
| RFC (RFE + RCS) | 11 | 2.71 | 4.88 | 0.51 | 2.30 |

Table 5: Quantitative comparison of different strategies for estimating the change in output features for DiT-XL/2 (Peebles & Xie, 2023) on ImageNet (Deng et al., 2009). All results are obtained using the first-order approximation ($m = 1$).

| Methods | NFC | FID2FC↓ | sFID2FC↓ | Latency (s)↓ |
|---|---|---|---|---|
| Linear | 14 | 0.73 | 3.40 | 2.85 |
| $w(t) = 0.8$ | 14 | 0.73 | 3.36 | 2.84 |
| $w(t) = 1.0$ | 14 | 0.57 | 2.77 | 2.84 |
| $w(t) = 1.2$ | 14 | 0.52 | 2.51 | 2.84 |
| RFE | 14 | 0.43 | 2.02 | 2.86 |
| Linear | 11 | 0.93 | 4.08 | 2.52 |
| $w(t) = 0.8$ | 11 | 0.94 | 4.17 | 2.51 |
| $w(t) = 1.0$ | 11 | 0.73 | 3.53 | 2.51 |
| $w(t) = 1.2$ | 11 | 0.67 | 3.24 | 2.52 |
| RFE | 11 | 0.55 | 2.55 | 2.52 |

Table 6: Quantitative comparison of different strategies for the cache scheduling using DiT-XL/2 (Peebles & Xie, 2023) on ImageNet (Deng et al., 2009). All results are obtained using the first-order approximation ($m = 1$).

| Methods | NFC | FID2FC↓ | sFID2FC↓ | Latency (s)↓ |
|---|---|---|---|---|
| Input distance | 14.03 | 0.50 | 2.53 | 2.62 |
| All modules | 14.02 | 0.37 | 1.85 | 2.79 |
| RCS | 14.00 | 0.36 | 1.88 | 2.63 |
| Input distance | 11.15 | 0.72 | 3.47 | 2.23 |
| All modules | 11.00 | 0.61 | 3.03 | 2.41 |
| RCS | 11.01 | 0.62 | 3.09 | 2.26 |

between the current and the latest cached input features, instead of exploiting the prediction error. We can see that RCS achieves stronger performance, suggesting that the prediction error is more reliable for forecasting-based methods than the simple distance measure. Second, we add the input prediction errors of all modules rather than using the first module only. RCS remains comparable to this variant, indicating that using the error from the first module is sufficient.

## 5 CONCLUSION

We have shown that output feature changes in DiTs are highly irregular across timesteps, while still maintaining a strong correlation with their corresponding inputs. Based on this, we have introduced a new caching framework, dubbed RFC, that leverages the input-output relationship for accurate feature prediction, consisting of two novel components, RFE and RCS. We have demonstrated that RFC achieves the state of the art on standard benchmarks, and further validated the effectiveness of each component through a detailed analysis.

ACKNOWLEDGEMENTS

This work was partly supported by IITP grant funded by the Korea government (MSIT) (No.RS-2022-00143524, Development of Fundamental Technology and Integrated Solution for Next-Generation Automatic Artificial Intelligence System, No.2022-0-00124, RS-2022-II220124, Development of Artificial Intelligence Technology for Self-Improving Competency-Aware Learning Capabilities), the KIST Institutional Program (Project No.2E33001-24-086), and Samsung Electronics Co., Ltd (IO240520-10013-01).

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

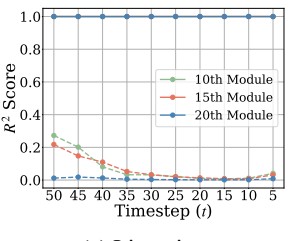 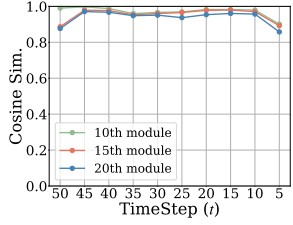 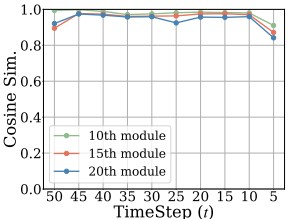

(a) Linearity.          (b) Input feature consistency.          (c) Output feature consistency.

Figure 5: Feature analyses using DiT-XL/2 (Peebles & Xie, 2023) on ImageNet (Deng et al., 2009). (a) A linearity analysis of diffusion features. The solid lines present the linearity between the input and output features, while the dashed lines show the linearity between the timesteps and output features. (b-c) The directional consistency of the input and output features.

In the Appendix, we first validate the conditions underlying Proposition 1 (Sec. A), followed by further analyses of the input-output feature relationship (Sec. B). We then provide more quantitative and qualitative results (Sec. C) with additional discussions (Sec. D). Finally, we present the limitations of RFC (Sec. E) and describe the usage of LLMs (Sec. F).

## A    CONDITIONS FOR PROPOSITION 1

Proposition 1 relies on two key conditions commonly observed in diffusion models: (1) a locally linear relationship between input and output features, and (2) directional consistency of feature changes across timesteps. The first condition is supported by the fact that feature changes between timesteps are typically small (Ma et al., 2024; Selvaraju et al., 2024; So et al., 2024), allowing for a local linear approximation by the Taylor's theorem. The second condition is grounded in empirical observations from prior works (Lv et al., 2025; Qiu et al., 2025), which observe that the direction of feature changes remains largely consistent throughout the denoising process. To further verify these, we provide additional empirical analyses on these conditions.

**Linearity.**   To validate the local linearity between the input and output features, we fit a linear model via least squares between the input and output features of the same module (*e.g.*, attention and MLP), using the features of 5 consecutive timesteps (*e.g.*, $t \sim (t-4)$). We show in Fig. 5(a) the average coefficient of determination ($R^2$) score of the fitted linear model. We can see that $R^2$ scores show consistently high values (*e.g.*, close to 1), demonstrating a strong linearity between input and output features. For comparison, we also compute the $R^2$ scores to quantify the linearity between the timesteps and their corresponding output features. We observe a substantial drop in $R^2$ scores, which indicates that the relationship between the output features and the timesteps is highly non-linear. This is because the magnitude of changes in the output features varies significantly, which limits the effectiveness of previous forecasting methods (Liu et al., 2025b; Lv et al., 2025; Qiu et al., 2025) (see Fig. 1(c-d)).

**Directional consistency.**   To validate the directional consistency of feature changes, we compute the pairwise cosine similarities of the feature differences $\Delta_k I(t - k)$ and $\Delta_k O(t - k)$, where $k$ ranges from 1 to 4. Specifically, we compute the difference vectors of features across timesteps (*e.g.*, $\Delta_k I(t-k) = I(t-k) - I(t)$ for $k = 1$ to 4), and measure the pairwise cosine similarities among them to assess how consistent their directions are over time. We then average these similarities to quantify the overall degree of directional consistency. As shown in Fig. 5(b-c), the average similarities remain consistently high across modules, indicating that the direction of feature change is preserved over time. These empirical observations align well with the findings of prior studies (Lv et al., 2025; Qiu et al., 2025).

## B    MORE ANALYSES

We have shown the strong correlation between the input and output features in the main paper. To further validate the generalizability of the analyses, we provide in Fig. 6 an additional analysis on the

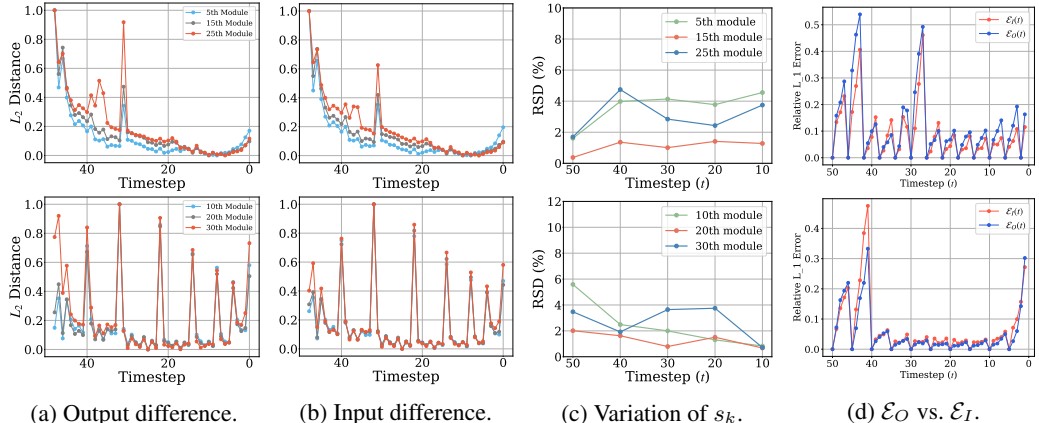

Figure 6: Analyses using FLUX.1 dev (Batifol et al., 2025) on DrawBench (Saharia et al., 2022) (upper row) and HunyuanVideo (Kong et al., 2024) on VBench (Huang et al., 2024) (lower row). (a-b) Min-max normalized $L_2$ distances of output and input features, measured between consecutive timesteps. (c) RSD of $s_k(t-k)$ in Eq. (5) with varying $t$. (d) Relative $L_1$ errors of the output and input features, *i.e.*, $\mathcal{E}_O(t)$ and $\mathcal{E}_I(t)$ in Eq. (21), respectively.

Table 7: Quantitative comparison under extremely high acceleration ratios for DiT-XL/2 (Peebles & Xie, 2023) on ImageNet (Deng et al., 2009). All results are obtained using the second-order approximation ($m=2$).

| Methods | NFC | FID↓ | sFID↓ | FID2FC↓ | sFID2FC↓ |
|---|---|---|---|---|---|
| Full-Compute | 50 | 2.32 | 4.32 | - | - |
| TaylorSeer (Liu et al., 2025b) | 6 | 13.57 | 13.19 | 10.68 | 15.19 |
| RFC | 6 | 4.31 | 5.39 | 1.83 | 4.80 |
| TaylorSeer (Liu et al., 2025b) | 5 | 45.81 | 27.64 | 42.29 | 31.46 |
| RFC | 5 | 5.43 | 6.29 | 2.81 | 6.13 |
| TaylorSeer (Liu et al., 2025b) | 4 | 160.85 | 135.55 | 157.53 | 137.65 |
| RFC | 4 | 8.65 | 7.14 | 5.71 | 8.78 |

Table 8: Quantitative comparison on distilled model, FLUX.1 schnell (Batifol et al., 2025) with 6 denoising steps, on DrawBench (Saharia et al., 2022). All results are obtained using the first-order approximation ($m=1$).

| Methods | NFC | PSNR↑ | SSIM↑ | LPIPS↓ |
|---|---|---|---|---|
| Full-Compute | 6 | - | 1.0000 | 0.0000 |
| Step reduction | 4 | 26.5377 | 0.8855 | 0.1164 |
| TaylorSeer (Liu et al., 2025b) | 4 | 29.3435 | 0.9242 | 0.0736 |
| RFC | 4 | **32.7436** | **0.9275** | **0.0635** |
| Step reduction | 3 | 25.0214 | 0.8580 | 0.1511 |
| TaylorSeer (Liu et al., 2025b) | 3 | 22.2134 | 0.7491 | 0.3687 |
| RFC | 3 | **27.1185** | **0.8931** | **0.0983** |

relationship between input and output using FLUX.1 dev (Batifol et al., 2025) on DrawBench (Saharia et al., 2022), and HunyuanVideo (Kong et al., 2024) on VBench (Huang et al., 2024). Similar to the findings in Figs. 1 and 2, we can see that the feature changes are irregular across timesteps, while those of input and output remain closely aligned with each other. We can also see that the ratio between the magnitude of changes in output and input features (*i.e.*, $s_k(t-k)$) is highly consistent along timesteps. Finally, we can see that the relative $L_1$ error of the output and input features, $\mathcal{E}_O(t)$ and $\mathcal{E}_I(t)$, respectively, are closely aligned with each other, demonstrating the strong correlation between the input and output features.

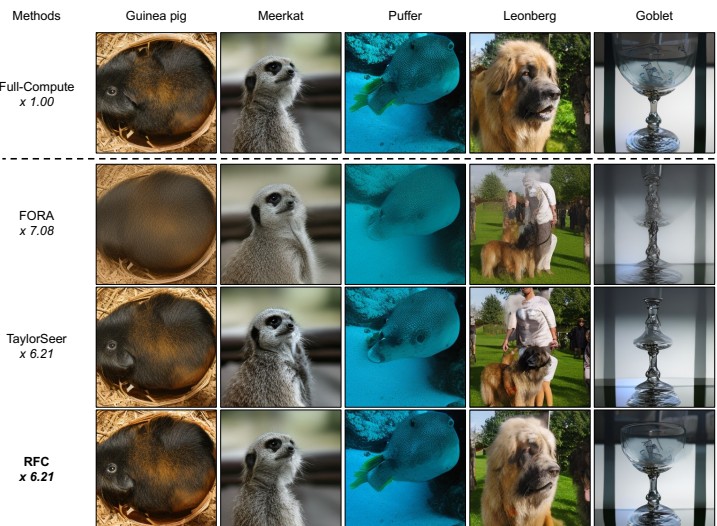

Figure 7: Qualitative comparisons of class-conditional image generation for DiT-XL/2 (Peebles & Xie, 2023) on ImageNet (Deng et al., 2009).

## C  MORE RESULTS

**Quantitative results on higher acceleration ratios.**  We show in Table 7 quantitative results using DiT-XL/2 (Peebles & Xie, 2023) on ImageNet (Deng et al., 2009) under extremely low full computation budgets (*i.e.*, NFC = 6, 5, 4). We can see that the performance of TaylorSeer (Liu et al., 2025b) degrades significantly. This indicates that relying solely on temporal extrapolation is insufficient when intervals between full computations are large. In contrast, RFC maintains strong performance by leveraging input–output correlations, which enables more accurate prediction even when feature changes become irregular and difficult to extrapolate.

**Quantitative results on distilled model.**  We show in Table 8 quantitative results on the distilled model, FLUX.1 schnell (Batifol et al., 2025), with 6 denoising steps. For the distilled models, the overall number of denoising steps is significantly smaller, which leads to larger changes in features between consecutive timesteps and makes feature prediction more difficult. Under this setting, we can see that TaylorSeer (Liu et al., 2025b) shows a notable performance drop, while RFC maintains consistently strong performance across all metrics. This demonstrates the effectiveness of leveraging the input–output relationship for stable and accurate feature prediction, even when the output features of distilled models highly fluctuate.

**More qualitative results.**  We provide in Figs. 7-9 additional qualitative results for class-conditional image, text-to-image, and text-to-video generation tasks. Consistent with the observations in Sec. 4.2, RFC produces higher-quality images and videos compared to those of state-of-the-art approaches (Selvaraju et al., 2024; Liu et al., 2025b). These results further validate the effectiveness and superiority of RFC.

## D  MORE DISCUSSIONS

**Details on choosing $\tau$.**  In our framework, $\tau$ is a key parameter in RCS that controls the trade-off between generation quality and computational efficiency. Specifically, a larger $\tau$ allows more prediction steps before triggering a full computation, thereby reducing the number of full computations and improving efficiency. This role is similar to adjusting $\delta$ in (Liu et al., 2025a) or $N$ in (Liu et al., 2025b; Selvaraju et al., 2024; Zou et al., 2024), where users can tune the parameter based on their desired efficiency level.

For fair comparison with other methods, we tune $\tau$ to match the target NFC. Specifically, we conduct a grid search over candidate $\tau$ values, generate 10 samples per setting, and measure the average NFC.

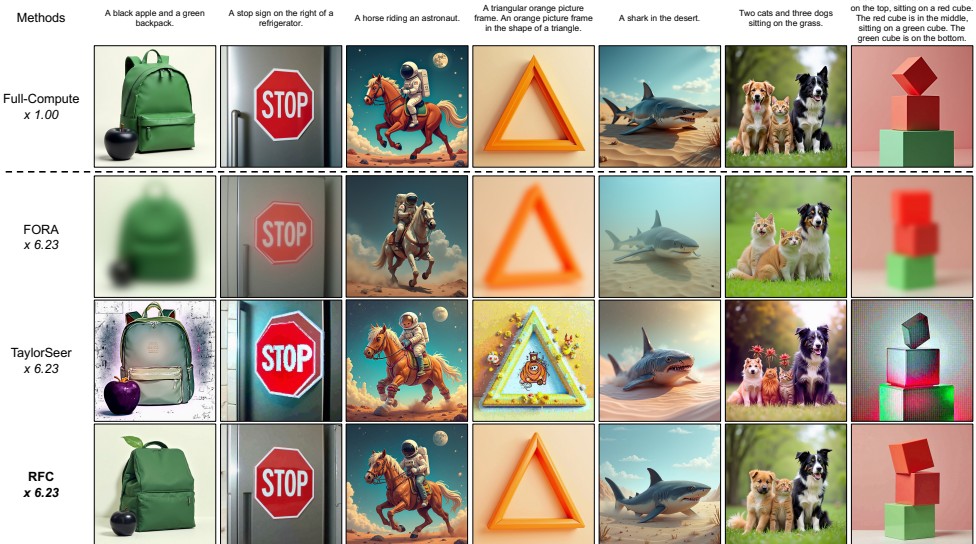

Figure 8: Qualitative comparisons of text-to-image generation for FLUX.1 dev (Batifol et al., 2025) on DrawBench (Saharia et al., 2022).

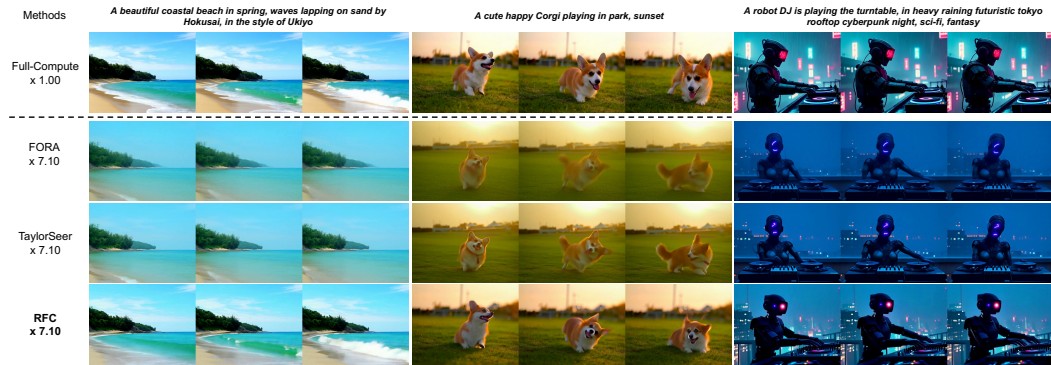

Figure 9: Qualitative comparisons of text-to-video generation for HunyuanVideo (Kong et al., 2024) on VBench (Huang et al., 2024).

We then fix the $\tau$ value that yields the desired NFC, and use this setting for reporting quantitative results. For detailed $\tau$ settings for Tables 1–3 are shown in Table 9. We also provide the trade-off of our approach with varying $\tau$ in Fig. 10. Please note that we have observed that NFC remains highly consistent across different samples, requiring no extensive tuning.

**Reliability of CLIP score.** RFC does not achieve the highest CLIP score in Table 2 due to the limitations of the CLIP score. The CLIP score mainly reflects coarse semantic alignment between the images and texts rather than detailed correctness, as demonstrated by prior work (Lin et al., 2024). Also, we can see in Table 2 that the CLIP score of TaylorSeer frequently outperforms fully computed counterparts, which shows its unreliability.

To further validate this, we show in Fig. 11 qualitative results where CLIP scores fail to reflect true generation quality. We can see that even when the generated images exhibit visual artifacts, they can still receive high CLIP scores when they roughly match the prompt. For instance, TaylorSeer (Liu et al., 2025b) generates a dog with a distorted face and a cat with unrealistic colors (first row), a clock where the number "9" is inaccurately shaped (second row), and an umbrella with unnatural textures (third row), but these examples show the highest CLIP scores.

Table 9: Settings of $\tau$ for various models.

| Model | Order ($m$) | $\tau$ | NFC |
|---|---|---|---|
| DiT-XL/2 (Peebles & Xie, 2023) | 2 | 0.19 | 14.01 |
| | | 0.38 | 11.01 |
| | | 0.48 | 10.04 |
| | | 0.88 | 8.02 |
| | | 1.20 | 7.04 |
| FLUX.1 dev (Batifol et al., 2025) | 1 | 0.30 | 14.02 |
| | | 1.10 | 8.00 |
| | 2 | 0.34 | 13.80 |
| | | 1.15 | 8.03 |
| HunyuanVideo (Kong et al., 2024) | 1 | 0.70 | 8.96 |
| | | 1.10 | 7.09 |

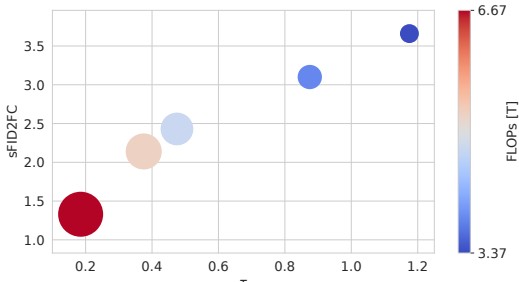

Figure 10: Trade-off between generation quality and computational cost on ImageNet (Deng et al., 2009) using DiT-XL/2 (Peebles & Xie, 2023) under varying values of $\tau$. The plot shows the change in performance (*i.e.*, sFID2FC) as $\tau$ increases, with the corresponding FLOPs indicated by the size and color of each circle.

Table 10: Quantitative results for layer-wise caching strategies on ImageNet (Deng et al., 2009) using DiT-XL/2 (Peebles & Xie, 2023). We reduce the frequency of full computations by half for the first (shallow skip) or last (deep skip) 7 blocks out of 28. We adjust $\tau$ to use more full computations for these variants to match FLOPs. All results are obtained using the second-order approximation ($m = 2$).

| Methods | NFC | FLOPs (T)↓ | FID↓ | sFID↓ | FID2FC↓ | sFID2FC↓ | IS↑ |
|---|---|---|---|---|---|---|---|
| Full-Compute | 50 | 23.74 | 2.32 | 4.32 | - | - | 241.25 |
| Shallow Skip | 16 | 6.66 | 3.09 | 4.65 | 0.75 | 2.61 | 220.00 |
| Deep Skip | 16 | 6.66 | 3.47 | 5.15 | 1.10 | 2.82 | 211.54 |
| RFC ($m = 2$) | 14 | 6.66 | 2.52 | 4.60 | 0.30 | 1.33 | 231.00 |
| Shallow Skip | 8 | 3.35 | 11.70 | 9.13 | 9.10 | 11.71 | 147.88 |
| Deep Skip | 8 | 3.35 | 10.20 | 6.22 | 6.95 | 6.59 | 149.10 |
| RFC ($m = 2$) | 7 | 3.35 | 3.40 | 5.21 | 1.03 | 3.66 | 215.39 |

**Layer-wise caching strategy.** We investigate whether skipping full computations for certain layers leads to a better trade-off between image quality and efficiency. Specifically, we consider two strategies: skipping full computations more often for (1) the shallow layers (*i.e.*, the first 7 blocks), or (2) the deep layers (*i.e.*, the last 7 blocks). In these settings, the selected layers perform fewer full-compute steps (*i.e.*, at half the frequency) than the remaining layers during the denoising process. We can see from Table 10 that both variants result in worse performance compared to our method, indicating that naive layer-wise skipping does not lead to better overall results. We believe, however, that exploring the strategies for layer-wise skipping could be a promising direction for future work.

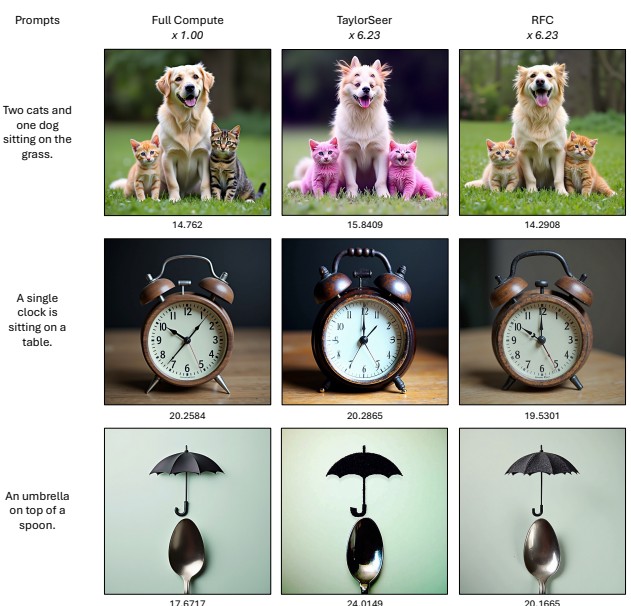

Figure 11: Clip scores of text-to-image generation for FLUX.1 dev (Batifol et al., 2025) on Draw-Bench (Saharia et al., 2022).

Table 11: Quantitative comparison of FORA (Selvaraju et al., 2024), TaylorSeer (Liu et al., 2025b), and RFC using the DDIM (Song et al., 2020) model trained on LSUN-bedroom (Yu et al., 2015) with 50 timesteps. We use the first-order approximation ($m = 1$).

| Methods | NFC | FID2FC↓ | sFID2FC↓ |
|---|---|---|---|
| FORA (Selvaraju et al., 2024) (N=5) | 11 | 78.27 | 69.84 |
| TaylorSeer (Liu et al., 2025b) (N=5) | 11 | 18.83 | 11.83 |
| RFC | 11 | **8.55** | **7.59** |
| FORA (Selvaraju et al., 2024) (N=6) | 10 | 103.48 | 89.17 |
| TaylorSeer (Liu et al., 2025b) (N=6) | 10 | 37.81 | 21.35 |
| RFC | 10 | **11.64** | **8.25** |

**Adaptability to U-Net architecture (Ronneberger et al., 2015).** Although we mainly focus on DiTs, RFC can also be applied to U-Net architectures. To validate its effectiveness, we compare in Table 11 RFC with prior methods (FORA (Selvaraju et al., 2024) and TaylorSeer (Liu et al., 2025b)) using the DDIM (Song et al., 2020) model trained on LSUN-bedroom (Yu et al., 2015). We can see that RFC significantly outperforms others, demonstrating its generality and architectural adaptability.

**Non-uniform intervals.** Our RCS dynamically determines when to perform full computations by monitoring the accumulated error in feature prediction. To investigate how this error varies across the denoising process, we show in Fig. 12 the average interval between two successive full-compute steps across 1,000 samples. We observe that the average interval between early full computations (i.e., close to noises) tends to be large, and the interval progressively shortens for the later timesteps (i.e., close to clean images). This indicates that prediction error accumulates more slowly at the beginning of the denoising process.

This phenomenon aligns with the generative process, where early stages form coarse, low-frequency structures with more stable feature dynamics, while later stages refine fine-grained, high-frequency details, involving more dynamic feature changes. RCS naturally adapts to this behavior by performing full computations less frequently during the early timesteps, when feature prediction is easier,

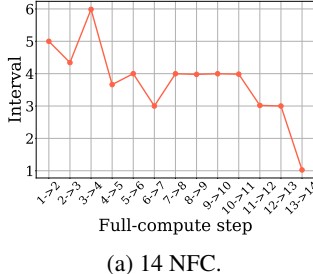 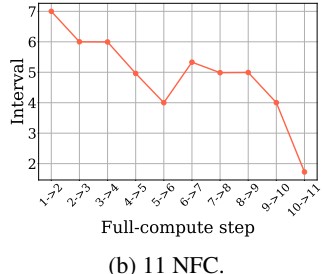

| (a) 14 NFC. | (b) 11 NFC. |

Figure 12: Analysis of non-uniform intervals scheduled by RCS using DiT-XL/2 (Peebles & Xie, 2023) on ImageNet (Deng et al., 2009). We visualize the average interval between full-compute steps for (a) 14 and (b) 11 NFC settings. $i \to i + 1$ at the x-axis denotes the $i$-th and $(i + 1)$-th full computation, and the y-axis shows the average interval. The results are averaged across 1,000 samples.

Table 12: Quantitative results of DiT-XL/2 (Peebles & Xie, 2023) on ImageNet (Deng et al., 2009), in terms of NFC, order in the Taylor expansion, GPU peak memory, FID2FC, and sFID2FC.

| Methods | NFC | Order | Peak Memory (MB) | FID2FC↓ | sFID2FC↓ |
|---|---|---|---|---|---|
| FORA (Selvaraju et al., 2024) | 10 | – | 3121 | 5.32 | 12.87 |
| TaylorSeer (Liu et al., 2025b) | 10 | 1 | 3207 | 1.42 | 6.17 |
| RFC | 10.04 | 1 | 3335 | 0.58 | 2.79 |
| TaylorSeer (Liu et al., 2025b) | 10 | 2 | 3333 | 1.05 | 4.86 |
| RFC | 10.02 | 2 | 3463 | 0.54 | 2.43 |

and increasing the frequency in the later timesteps as prediction becomes more challenging, thereby optimizing the efficiency-quality trade-off.

# E LIMITATION

Table 13: Time overhead of RFC components compared to the baseline (TaylorSeer (Liu et al., 2025b)) using DiT-XL/2 (Peebles & Xie, 2023) on ImageNet (Deng et al., 2009) with 14 NFCs. Percentages in parentheses indicate the time increase relative to the baseline.

| Methods | Time (Overhead %) |
|---|---|
| Baseline (TaylorSeer (Liu et al., 2025b)) | 2.840s |
| Input feature computation | + 0.019s (0.67%) |
| Input feature prediction | + 0.002s (0.07%) |

RFC effectively improves the generation quality but incurs additional memory costs to store input features, similar to TaylorSeer (Liu et al., 2025b), which stores more output features for higher-order approximations. Notably, as shown in Table 12, RFC generally achieves greater improvements in the generation quality than increasing the approximation order by one in TaylorSeer, while requiring a comparable amount of memory, highlighting the effectiveness of our approach.

As shown in Table 13, RFC incurs a slight time overhead compared to the baseline. Compared to the baseline, RFC needs (1) input feature computation for RFE and RCS, which requires lightweight operations (*e.g.*, LayerNorm, scaling and shifting); and (2) input feature prediction using the Taylor expansion for RCS. To reduce the cost of input feature prediction in RCS, we have demonstrated in Table 6 that computing the prediction error from only the first module, rather than all modules (*e.g.*, 56 modules in DiT-XL/2), is sufficient. For instance, under the 11 NFC setting, computing prediction error across all modules increases runtime by approximately 7%, while yielding only marginal improvement in performance (0.62 vs. 0.61 in terms of FID2FC). We conjecture that this is because errors in early modules affect subsequent modules, suggesting that the prediction error of the first module can be a reliable indicator of the overall prediction error. To further reduce the cost of input feature computation, a promising direction for future work could be to explore a hybrid strategy

that applies RFE only to a subset of important modules, while using the baseline method for the remaining ones.

There are some failure cases in RFC, where the generated image does not align well with the prompt. This typically occurs when full computations fail to capture the prompt correctly, for example, as shown in the third row of Fig. 8, where an astronaut is riding a horse, instead of a horse riding an astronaut. This is because RFC accurately approximates the full computations, it also reproduces such errors.

## F  LLM USAGE

We used a large language model (LLM) solely for proofreading and polishing the manuscript. The model was not involved in generating research ideas, designing methods, conducting experiments, or interpreting results.

