# OpenReview forum: "Relational Feature Caching for Accelerating Diffusion Transformers"
_ICLR.cc/2026/Conference — ICLR 2026 Poster_

### Official Review · Reviewer_xGHL · 2025-10-27

**Soundness:** 3
**Presentation:** 2
**Contribution:** 3
**Rating:** 6
**Confidence:** 5

**Summary:**

This paper proposes an improvement of TayloSeer, motivated by the highly correlated magnitudes of change of input and output features. It uses the change of input features to estimate the change of output, therefore reducing the computation to a large extent. On both text-to-image and text-to-video tasks, RFC manages to achieve better or comparable performances against baselines. Visual results from the paper also support the effectiveness of this method.

**Strengths:**

1. The motivation of this paper is clear. The empirical results in Figures 1 and 2 serve as strong evidence for the proposed method.
2. The discussion of related work is thorough, further enhancing the novelty of this paper.
3. The experimental analysis is comprehensive. The authors validate their method not just on one model or task, but across three distinct, large-scale generative tasks (class-conditional, T2I, T2V) using modern, powerful models.

**Weaknesses:**

Please see the weakness of the method and experiments in the questions.

Here are some weaknesses in the presentation:
1. The related work section is a bit hard to read with the long paragraph. I suggest the author reorganize this section to improve the readability.
2. I suggest using the full name of RFE and RCS as the paragraph header.

**Questions:**

1. How does RFC perform on distilled models?
2. What is the recomputation rate under different parameter settings? In other words, is $\tau$ hard to tune for different DiT models? It seems this value differs across the DiT models tested.
3. The empirical results (Figures 1 and 2) are obtained on ImageNet and DiT, which are relatively simple. Are the findings the same in a different case? For example, it might be more convincing to also show empirical results on FLUX.1 dev or HunyuanVideo.
4. RFC seems to be slower than the baseline. Where does the main overhead come from, and did the author think about how to reduce it?
5. Did the author study whether shallow layers / early timesteps should be skipped or not to achieve better results?

---

> ### Author Response · Authors · 2025-11-20
> **Response to Reviewer xGHL [1/3]**
>
> We sincerely appreciate your thoughtful and constructive feedback on our manuscript. Below, we provide detailed responses to each of your comments. Major revisions in the updated manuscript are highlighted in blue for clarity.
>
>
> ----------
> > **W1 : The related work section is a bit hard to read with the long paragraph.**
>
> - Thank you for your comment. We have reorganized the related work section to improve clarity and readability.
>
>
> ----------
> > **W2 : I suggest using the full name of RFE and RCS as the paragraph header.**
>
> - We appreciate the suggestion. Following your advice, we have revised the paragraph headers to use the full names.
>
>
> ----------
> > **Q1 : How does RFC perform on distilled models?**
>
> - Thank you for your insightful question. We show in Table R1 (or Table 8 in Sec. C) quantitative results on the distilled FLUX.1 schnell model with 6 denoising steps. For the distilled models, the overall number of denoising steps is significantly smaller, which leads to larger changes in features between consecutive timesteps and makes feature prediction more difficult. Under this setting, we can see that TaylorSeer shows a notable performance drop, while RFC maintains consistently strong performance across all metrics. This demonstrates the effectiveness of leveraging the input–output relationship for stable and accurate feature prediction, even when the output feature trajectories of distilled models exhibit significant fluctuations. We have added these results in Sec. C.
>
>
> **Table R1** : Quantitative comparison on distilled model, FLUX.1 schnell, on DrawBench. All results are obtained using the first-order approximation ($m=1$).
>
> \begin{array}{l c c c c}
> \hline
> \textbf{Methods} & \textbf{NFC} & \textbf{PSNR}\uparrow & \textbf{SSIM}\uparrow & \textbf{LPIPS}\downarrow \newline
> \hline
> \text{Full-Compute} & 6 & - & 1.0000 & 0.0000 \newline
> \hline
> \text{Step reduction} & 4 & 26.5377 & 0.8855 & 0.1164 \newline
> \text{TaylorSeer} & 4 & 29.3435 & 0.9242 & 0.0736 \newline
> \text{RFC} & 4 & \textbf{32.7436} & \textbf{0.9275} & \textbf{0.0635} \newline
> \hline
> \text{Step reduction} & 3 & 25.0214 & 0.8580 & 0.1511 \newline
> \text{TaylorSeer} & 3 & 22.2134 & 0.7491 & 0.3687 \newline
> \text{RFC} & 3 & \textbf{27.1185} & \textbf{0.8931} & \textbf{0.0983} \newline
> \hline
> \end{array}
>
> ----------
> > **Q2 : Is $\tau$ hard to tune for different DiT models?**
>
> - Thank you for the question. We have included detailed information on how $\tau$ is adjusted in Sec. D. In our framework, $\tau$ in RCS controls the trade-off between generation quality and computational efficiency. A larger $\tau$ allows more prediction steps before triggering a full computation, which leads to fewer full-compute operations and higher efficiency, similar to the role of $\delta$ in [1] or $N$ in [2].
>
> - To ensure a fair comparison across methods, we tune $\tau$ to match the target NFC. Specifically, we conduct a grid search over candidate $\tau$ values, generate 10 samples per setting, and measure the average NFC. We then **fix** the $\tau$ value that yields the desired NFC, and use this setting for reporting quantitative results. We have observed that NFC remains highly consistent across different samples, requiring no extensive tuning.
>
>  - A summary of the chosen $\tau$ values for the main experiments can be found in Table R2 (or Table 9 of Sec. D), and we visualize the quality-efficiency trade-off under different $\tau$ settings in Fig. 10 of Sec. D.
>
>
> [1] Timestep embedding tells: It’s time to cache for video diffusion model, CVPR 2025
>
> [2] From reusing to forecasting: Accelerating diffusion models with taylorseers, ICCV 2025
>
> **Table R2** : Settings of $\tau$ for various models.
>
> \begin{array}{l c c c}
> \hline
> \textbf{Model} & \textbf{Order } (m) & \tau & \textbf{NFC} \newline
> \hline
> \text{DiT-XL/2} & 2 & 0.19 & 14.01 \newline
> & 2& 0.38 & 11.01 \newline
> & 2& 0.48 & 10.04 \newline
> & 2& 0.88 & 8.02 \newline
> & 2& 1.20 & 7.04 \newline
> \hline
> \text{FLUX.1 dev} & 1 & 0.30 & 14.02 \newline
> & 1& 1.10 & 8.00 \newline
> & 2 & 0.34 & 13.80 \newline
> & 2& 1.15 & 8.03 \newline
> \hline
> \text{HunyuanVideo} & 1 & 0.70 & 8.96 \newline
> & 1& 1.10 & 7.09 \newline
> \hline
> \end{array}

---

> ### Author Response · Authors · 2025-11-20
> **Response to Reviewer xGHL [2/3]**
>
> > **Q3 : It might be more convincing to also show empirical results of Figs. 1 and 2 on FLUX.1 dev or HunyuanVideo.**
>
> - Thank you for the feedback. The empirical analysis on FLUX.1 dev was already included in Fig. 6 of Sec. B, and we have further added the same analysis on HunyuanVideo in the same figure. Similar to the findings in DiT-XL/2, we can see that although the magnitude of feature changes differs significantly across timesteps, the input and output features are closely aligned throughout the sampling trajectory. Moreover, the ratio between the magnitudes of changes in input and output features (i.e., $s_k(t-k)$) stays highly consistent, and the prediction errors for the two features are nearly identical. This demonstrates that the strong input–output correlation holds across modern large-scale text-to-image and video diffusion models, highlighting the generalizability and robustness of our analysis across diverse architectures and modalities.
>
> -------
>
> > **Q4 : RFC seems to be slower than the baseline. Where does the main overhead come from, and did the author think about how to reduce it?**
>
> - Thank you for your question. As shown in Table R3, RFC incurs a slight time overhead compared to the baseline. Compared to the baseline, RFC needs (1) input feature computation for RFE and RCS, which requires lightweight operations (e.g., LayerNorm, scaling and shifting); and (2) input feature prediction using the Taylor expansion for RCS. To reduce the cost of input feature prediction in RCS, we have demonstrated in Table 6 that computing the prediction error only at the first module, rather than all modules (e.g., 56 modules in DiT-XL/2), is sufficient. For instance, under the 11 NFC setting, computing prediction error across all modules increases runtime by approximately 7%, while yielding only marginal improvement in performance (0.62 vs. 0.61 in terms of FID2FC). We conjecture that this is because errors in early modules affect subsequent modules, suggesting that the prediction error of the first module can be a reliable indicator of the overall prediction error. To further reduce the cost of input feature computation, a promising direction for future work could be to explore a hybrid strategy that applies RFE only to a subset of important modules, while using the baseline method for the remaining ones. We have added the discussion of the time overhead of our framework in Sec. E.
>
>
> **Table R3** : Time overhead of RFC components compared to the baseline (TaylorSeer) using DiT-XL/2 on ImageNet with 14 NFC. Percentages in parentheses indicate the time increase relative to the baseline.
>
> \begin{array}{l l}
> \hline
> \textbf{Methods} & \textbf{Time (Overhead \\%)} \newline
> \hline
> \text{Baseline (TaylorSeer)} & \text{2.840s} \newline
> \hline
> \text{Input feature computation} & \text{+ 0.019s (0.67 \\%)} \newline
> \text{Input feature prediction} & \text{+ 0.002s (0.07 \\%)} \newline
> \hline
> \end{array}

---

> ### Author Response · Authors · 2025-11-20
> **Response to Reviewer xGHL [3/3]**
>
> > **Q5 : Did the author study whether shallow layers / early timesteps should be skipped or not to achieve better results?**
>
> - Following your question,  we have investigated whether skipping full computations for certain layers leads to a better trade-off between image quality and efficiency. Specifically, we have considered two strategies: skipping full computations more often for (1) the shallow layers (i.e., the first 7 blocks), or (2) the deep layers (i.e., the last 7 blocks). In these settings, the selected layers perform fewer full-compute steps (i.e., at half the frequency) than the remaining layers during the denoising process. We can see from Table R4 (or Table 10 of Sec. D) that both variants result in worse performance compared to our method, indicating that naive layer-wise skipping does not lead to better overall results. We believe, however, that exploring meticulous strategies for layer-wise skipping would be a promising direction for future work.
>
>
> - We have also investigated which timesteps are more important to achieve better performance by observing how our RCS performs full computations. To this end, we show in Fig. 12 of Sec. D the average interval between successive full-compute steps across 1,000 samples. We can see that the average interval between early full computations (i.e., close to noises) tends to be large, and the interval progressively shortens for the later timesteps (i.e., close to clean images). This suggests that prediction errors accumulate more slowly in early stages, making them easier to approximate and thus more suitable for skipping.
>
>
> **Table R4** : Quantitative results for layer-wise caching strategies on ImageNet using DiT-XL/2. We reduce the frequency of full computations by half for the first (shallow skip) or last (deep skip) 7 blocks out of 28. We adjust $\tau$ to use more full computations for these variants to match FLOPs. All results are obtained using the second-order approximation ($m=2$).
>
> \begin{array}{l c c c c c c}
> \hline
> \textbf{Methods} & \textbf{NFC} & \textbf{FID}\downarrow & \textbf{sFID}\downarrow & \textbf{FID2FC}\downarrow & \textbf{sFID2FC}\downarrow & \textbf{IS}\uparrow \newline
> \hline
> \text{Full-Compute} & 50 & 2.32 & 4.32 & - & - & 241.25 \newline
> \hline
> \text{Shallow Skip} & 16 & 3.09 & 4.65 & 0.75 & 2.61 & 220.00 \newline
> \text{Deep Skip} & 16 & 3.47 & 5.15 & 1.10 & 2.82 & 211.54 \newline
> \text{RFC } (m=2) & 14 & \textbf{2.52} & \textbf{4.60} & \textbf{0.30} & \textbf{1.33} & \textbf{231.00} \newline
> \hline
> \text{Shallow Skip} & 8 & 11.70 & 9.13 & 9.10 & 11.71 & 147.88 \newline
> \text{Deep Skip} & 8 & 10.20 & 6.22 & 6.95 & 6.59 & 149.10 \newline
> \text{RFC } (m=2) & 7 & \textbf{3.40} & \textbf{5.21} & \textbf{1.03} & \textbf{3.66} & \textbf{215.39} \newline
> \hline
> \end{array}

---

> ### Comment · Reviewer_xGHL · 2025-11-26
>
> Thanks for the detailed response. My concerns have been mostly addressed. I will maintain my current score.

---

> > ### Author Response · Authors · 2025-11-27
> > **Response to Reviewer xGHL**
> >
> > Thank you for carefully considering our rebuttal. We are pleased to hear that it addressed your concerns. If you have any further questions or suggestions, please feel free to let us know.

---

### Official Review · Reviewer_h2W6 · 2025-10-28

**Soundness:** 3
**Presentation:** 3
**Contribution:** 3
**Rating:** 6
**Confidence:** 3

**Summary:**

This paper proposes Relational Feature Caching (RFC) to accelerate Diffusion Transformers by exploiting correlations between input and output features, rather than relying solely on temporal extrapolation as in prior methods like TaylorSeer. The approach introduces Relational Feature Estimation (RFE) to predict output changes from input variations and Relational Cache Scheduling (RCS) to adaptively trigger full computations based on estimated errors. Experiments on image, text-to-image, and video generation tasks show consistent improvements over existing caching methods in both quality and efficiency.

**Strengths:**

- Clear motivation with strong empirical evidence showing the limitation of purely temporal forecasting.
- Simple yet effective design—RFE and RCS are well-justified and complementary.
- Comprehensive experiments across multiple diffusion models and tasks.

**Weaknesses:**

- In Table 2, TaylorSeer achieves the highest CLIP scores; the paper should discuss why RFC does not consistently outperform it on semantic alignment metrics.
- The paper could analyze RFC’s applicability to U-Net–based diffusion models to better demonstrate generality and architectural adaptability.

**Questions:**

- While RFC significantly accelerates DiTs, what happens at higher acceleration ratios (larger N)? A discussion on generation quality degradation at extreme speedups would help readers assess its robustness.

---

> ### Author Response · Authors · 2025-11-20
> **Response to Reviewer h2W6**
>
> We sincerely appreciate your thoughtful and constructive feedback on our manuscript. Below, we provide detailed responses to each of your comments. Major revisions in the updated manuscript are highlighted in blue for clarity.
>
> ----------
> > **W1 : TaylorSeer achieves the highest CLIP scores.**
>
> - Thank you for the insightful comment. We’d like to clarify that this is due to the limitations of the CLIP metric. It mainly reflects coarse semantic alignment between the images and texts rather than detailed correctness, as demonstrated by prior work [1]. Also, we can see in Table 2 that the CLIP score of TaylorSeer frequently outperforms fully computed counterparts, which shows its unreliability. We have added examples where the CLIP metric is counterintuitive to human judgment in Fig. 11 of Sec. D. We have observed that overly simplified and unrealistic images often receive higher CLIP scores. Please see Sec. D for more discussions.
>
>
> [1] Evaluating text-to-visual generation with image-to-text generation, ECCV 2024
>
>
> ----------
> > **W2 : RFC’s applicability to the U–Net–based diffusion model is not demonstrated.**
>
> - Thank you for the valuable suggestion. RFC is compatible with U-Net-based models by predicting output features of each convolutional layer. To validate this, we show in Table R1 (or Table 11 of Sec. D) the quantitative results of FORA, TaylorSeer, and RFC using the DDIM model trained on the LSUN-bedroom dataset. We can see that RFC consistently achieves the best performance across varying NFCs, demonstrating its generality and architectural adaptability. We have added the results in Table 11 of Sec. D.
>
> **Table R1** : Quantitative comparison of FORA, TaylorSeer, and RFC using the DDIM model trained on LSUN-bedroom with 50 timesteps. We use the first-order approximation ($m=1$).
>
> \begin{array}{l c c c}
> \hline
> \textbf{Methods} & \textbf{NFC} & \textbf{FID2FC}\downarrow & \textbf{sFID2FC}\downarrow \newline
> \hline
> \text{FORA (N=5)} & 11 & 78.27 & 69.84 \newline
> \text{TaylorSeer (N=5)} & 11 & 18.83 & 11.83 \newline
> \text{RFC} & 11 & \textbf{8.55} & \textbf{7.59} \newline
> \hline
> \text{FORA (N=6)} & 10 & 103.48 & 89.17 \newline
> \text{TaylorSeer (N=6)} & 10 & 37.81 & 21.35 \newline
> \text{RFC} & 10 & \textbf{11.64} & \textbf{8.25} \newline
> \hline
> \end{array}
>
>
>
> ----------
> > **Q1 : What happens at higher acceleration ratios?**
>
> - Thank you for the suggestion. We show in Table R2 (or Table 7 in Sec. C) quantitative results under extremely high acceleration ratios (low NFCs).  We can see that the performance of TaylorSeer degrades significantly. This indicates that relying solely on temporal extrapolation is insufficient when intervals between full computations are large. In contrast, RFC maintains strong performance by leveraging input–output correlations, which enables more accurate prediction even when feature changes become highly irregular.
>
>
> **Table R2** : Quantitative comparison under extremely high acceleration ratios for DiT-XL/2 on ImageNet. All results are obtained using the second-order approximation ($m=2$).
>
> \begin{array}{l c c c c c}
> \hline
> \textbf{Methods} & \textbf{NFC} & \textbf{FID}\downarrow & \textbf{sFID}\downarrow & \textbf{FID2FC}\downarrow & \textbf{sFID2FC}\downarrow \newline
> \hline
> \text{Full-Compute} & 50 & 2.32 & 4.32 & - & - \newline
> \hline
> \text{TaylorSeer} & 6 & 13.57 & 13.19 & 10.68 & 15.19 \newline
> \text{RFC} & 6 & \textbf{4.31} & \textbf{5.39} & \textbf{1.83} & \textbf{4.80} \newline
> \hline
> \text{TaylorSeer} & 5 & 45.81 & 27.64 & 42.29 & 31.46 \newline
> \text{RFC} & 5 & \textbf{5.43} & \textbf{6.29} & \textbf{2.81} & \textbf{6.13} \newline
> \hline
> \text{TaylorSeer} & 4 & 160.85 & 135.55 & 157.53 & 137.65 \newline
> \text{RFC} & 4 & \textbf{8.65} & \textbf{7.14} & \textbf{5.71} & \textbf{8.78} \newline
> \hline
> \end{array}

---

### Official Review · Reviewer_6daa · 2025-10-31

**Soundness:** 3
**Presentation:** 2
**Contribution:** 3
**Rating:** 6
**Confidence:** 3

**Summary:**

Objective: Accelerate Diffusion Transformers by improving feature caching accuracy. The paper identifies that forecast-based caches suffer from large errors due to irregular output changes and strong input–output correlations in modules. It proposes Relational Feature Caching (RFC) with two components: Relational Feature Estimation (RFE) to predict output-change magnitudes from inputs, and Relational Cache Scheduling (RCS) to estimate prediction error from inputs and trigger full computation only when errors are likely high. Experiments across multiple DiT models show consistent, significant improvements over temporal extrapolation baselines, with planned code release upon acceptance.

**Strengths:**

This paper studies an important topic of feature caching, which is critical for optimizing the efficiency of diffusion transformers.

The identification of input–output correlation as a predictor of output changes is insightful and well supported by the empirical analysis.

The proposed RFC showcase on multiple metrics and settings, demonstrating a comparable to superior performance. The authors also present qualitative results, which are promising.

**Weaknesses:**

The organization of the manuscript needs improvement. There is overlapping and duplicate content across the first three sections. It may be better to defer the detailed discussion of related work to a later section and to reorganize Section 2 and Section 3.1.

While the empirical correlation between inputs and outputs is demonstrated, the paper offers little formal analysis explaining why this correlation should hold across arbitrary architectures or datasets.

**Questions:**

line 302-304, "For a fair comparison, we reproduce the results of state-of-the-art methods using the official source codes, and adjust the threshold τ in Eq. (13) to ensure that the average number of full computations (NFC) matches that of other methods." can you elaborate more on how to adjust the threshold tau?

---

> ### Author Response · Authors · 2025-11-20
> **Response to Reviewer 6daa**
>
> We sincerely appreciate your thoughtful and constructive feedback on our manuscript. Below, we provide detailed responses to each of your comments. Major revisions in the updated manuscript are highlighted in blue for clarity.
>
> ----------
> > **W1 : Section 2 and Section 3.1 need to be reorganized.**
>
> - Thank you for the helpful suggestion. Based on your feedback, we have reorganized the sections to improve clarity and removed redundant content. Specifically, we have deferred the detailed discussion of related work to a later section and restructured Sec. 2 and 3.1 to streamline the presentation of our method.
>
> ----------
> > **W2 : Lack of formal analysis of input-output correlation.**
>
> - Thank you for the insightful comment. To address this, we have reorganized and added Proposition 1 to the main paper, along with a detailed formal proof, which theoretically explains the input–output correlation (i.e., consistency of $s_k(t-k)$).
>
> - Proposition 1 shows that $s_k(t-k)$ is highly consistent under two general conditions in diffusion models: (1) local linearity between input and output features, and (2) directional consistency of feature changes across timesteps, where both are well supported by prior studies [1, 2, 3]. Please also see Proposition 1 in Sec. 3.2 and analysis in Sec. A.
>
> - In addition, we have extended our empirical analysis beyond DiT-XL/2 and FLUX.1-dev by including results from HunyuanVideo (Fig. 6 of Sec. B), showing that the input–output correlation remains consistent across different architectures and datasets. We believe this strengthens the generality of our approach.
>
> [1] DeepCache: Accelerating diffusion models for free, CVPR 2024
>
> [2] FasterCache: Training-free video diffusion model acceleration with high quality, ICLR 2025
>
> [3] Accelerating diffusion transformer via gradient-optimized cache, ICCV 2025
>
>
> ----------
> > **Q1 How to adjust $\tau$?**
>
> - Thank you for the question. We have included detailed information on how $\tau$ is adjusted in Sec. D. In our framework, $\tau$ in RCS controls the trade-off between generation quality and computational efficiency. A larger $\tau$ allows more prediction steps before triggering a full computation, which leads to fewer full-compute operations and higher efficiency, similar to the role of $\delta$ in [4] or $N$ in [5].
>
> - To ensure a fair comparison across methods, we tune $\tau$ to match the target NFC. Specifically, we conduct a grid search over candidate $\tau$ values, generate 10 samples per setting, and measure the average NFC. We then **fix** the $\tau$ value that yields the desired NFC, and use this setting for reporting quantitative results.
>
>  - A summary of the chosen $\tau$ values for the main experiments can be found in Table R1 (or Table 9 of Sec. D), and we visualize the quality-efficiency trade-off under different $\tau$ settings in Fig. 10 of Sec. D.
>
> [4] Timestep embedding tells: It’s time to cache for video diffusion model, CVPR 2025
>
> [5] From reusing to forecasting: Accelerating diffusion models with taylorseers, ICCV 2025
>
>
> **Table R1** : Settings of $\tau$ for various models.
>
> \begin{array}{l c c c}
> \hline
> \textbf{Model} & \textbf{Order } (m) & \tau & \textbf{NFC} \newline
> \hline
> \text{DiT-XL/2} & 2 & 0.19 & 14.01 \newline
> & 2& 0.38 & 11.01 \newline
> & 2& 0.48 & 10.04 \newline
> & 2& 0.88 & 8.02 \newline
> & 2& 1.20 & 7.04 \newline
> \hline
> \text{FLUX.1 dev} & 1 & 0.30 & 14.02 \newline
> & 1& 1.10 & 8.00 \newline
> & 2 & 0.34 & 13.80 \newline
> & 2& 1.15 & 8.03 \newline
> \hline
> \text{HunyuanVideo} & 1 & 0.70 & 8.96 \newline
> & 1& 1.10 & 7.09 \newline
> \hline
> \end{array}

---

### Official Review · Reviewer_gkp8 · 2025-11-01

**Soundness:** 3
**Presentation:** 2
**Contribution:** 3
**Rating:** 6
**Confidence:** 2

**Summary:**

This paper addresses inefficiencies in diffusion transformers (DiTs) by advancing feature caching techniques used to accelerate inference. Prior approaches speed up computation by temporally extrapolating and reusing output features of expensive modules, but these methods can incur significant prediction errors. Through detailed analysis, the authors find that such errors arise from irregular feature magnitude changes and that there is a strong correlation between a module’s input and its output features. Building on these insights, they introduce Relational Feature Caching (RFC), a novel framework that leverages the relationship between inputs and outputs to improve feature prediction accuracy. RFC includes two key components: Relational Feature Estimation (RFE), which uses input features to predict output changes more reliably, and Relational Cache Scheduling (RCS), which estimates likely prediction errors from inputs and recomputes outputs only when large errors are expected. Experiments on various DiT models show that RFC consistently outperforms earlier techniques, significantly improving efficiency and accuracy.

**Strengths:**

This work introduces novel components—Relational Feature Estimation (RFE) and Relational Cache Scheduling (RCS)—that have not been explored in prior work. It utilizes input–output relationships, enabling a more accurate prediction of output features. The paper clearly articulates the motivation, challenges, and contributions, and provides a general framework that could inspire further research for DiTs.

**Weaknesses:**

Clarity of Mathematical Formulation (RELATIONAL FEATURE CACHING section):

The presentation of equations in the RELATIONAL FEATURE CACHING section lacks clarity, making it difficult for readers to follow the mathematical foundations of the approach. For instance, the connection between the two components, RFE and RCS, is not clear at the beginning, and the logical flow from one equation to the next is not always well-motivated.

Actionable suggestion: Including intuitive descriptions or intermediary steps (potentially with illustrative diagrams or simplified toy examples) would help clarify the input-output relationship modeling and the estimation process.

Experimental Evaluation Coverage:

While the experiments show consistent gains across several DiT models, the paper primarily focuses on performance improvements and does not fully explore scenarios where the method may fail or be less effective (e.g., with particularly noisy or weak input-output correlations).
Actionable suggestion: Include ablation studies or failure case analysis to identify situations where the relational estimation approach may struggle or need further modification. Examining a broader range of conditions, such as varying degrees of input-output correlation, would provide a more comprehensive understanding of the method's robustness. For Table 4, provide a brief written summary in the main text to guide the reader through the table and highlight the most important results.

**Questions:**

How does the model perform in terms of efficiency when the RCS component has a different scheduling policy?

Is there a clear trade-off pattern for the method?

---

> ### Author Response · Authors · 2025-11-20
> **Response to Reviewer gkp8**
>
> We sincerely appreciate your thoughtful and constructive feedback on our manuscript. Below, we provide detailed responses to each of your comments. Major revisions in the updated manuscript are highlighted in blue for clarity.
>
> ---------------
> > **W1 : The presentation of equations in Sec. 3.2 lacks clarity.**
>
> - Thank you for pointing out the clarity issues in the presentation. We have revised Sec. 3 thoroughly based on your suggestion. Specifically, we have added the description of the connection between RFE and RCS at the beginning (Lines 193-196), and we have strengthened the logical flow between the equations by adding intermediary steps (Lines 245-253).
>
>
>
>
> ---------------
> >**W2 : Explore the scenarios where the method may fail or be less effective. Also, provide a brief summary of Table 4 in the main text.**
>
>
> - We agree that it is important to analyze the limitations and explore failure cases of our work. RFC relies on the strong correlation between input and output features, which generally holds across diverse models and datasets. However, we acknowledge that its effectiveness can be challenged under extremely low NFC settings, where full-compute steps are too sparse. In such cases, the magnitude of feature changes between cached steps becomes large, which can weaken the local linearity assumption underlying Proposition 1 and reduce prediction accuracy.
>
> - To investigate this, we show in Table R1 an ablation study under extremely low NFC settings (4, 5, or 6), where prediction becomes significantly more difficult. We can see from the results that while the performance of all methods degrades under such extreme conditions, RFC consistently outperforms TaylorSeer, which relies solely on temporal extrapolation. This suggests that even when the input–output correlation is less stable, it remains a more reliable predictor than simply extrapolating features over time, which struggles to capture irregular feature dynamics over long intervals. We have added these results in Sec. C. Please also refer to Sec. E for additional analyses on the limitations of our method.
>
> - We also have strengthened a summary of Table 4 in the main text to guide the reader through the table and highlight the most important results (Lines 468-473).
>
>
> **Table R1** : Quantitative comparison under extremely high acceleration ratios for DiT-XL/2 on ImageNet. All results are obtained using the second-order approximation ($m=2$).
>
> \begin{array}{l c c c c c}
> \hline
> \textbf{Methods} & \textbf{NFC} & \textbf{FID}\downarrow & \textbf{sFID}\downarrow & \textbf{FID2FC}\downarrow & \textbf{sFID2FC}\downarrow \newline
> \hline
> \text{Full-Compute} & 50 & 2.32 & 4.32 & - & - \newline
> \hline
> \text{TaylorSeer} & 6 & 13.57 & 13.19 & 10.68 & 15.19 \newline
> \text{RFC} & 6 & \textbf{4.31} & \textbf{5.39} & \textbf{1.83} & \textbf{4.80} \newline
> \hline
> \text{TaylorSeer} & 5 & 45.81 & 27.64 & 42.29 & 31.46 \newline
> \text{RFC} & 5 & \textbf{5.43} & \textbf{6.29} & \textbf{2.81} & \textbf{6.13} \newline
> \hline
> \text{TaylorSeer} & 4 & 160.85 & 135.55 & 157.53 & 137.65 \newline
> \text{RFC} & 4 & \textbf{8.65} & \textbf{7.14} & \textbf{5.71} & \textbf{8.78} \newline
> \hline
> \end{array}
>
>
>
> ---------------
> > **Q1 : How does the model perform in terms of efficiency when the RCS component has a different scheduling policy?**
>
>
> - Thank you for your suggestion. We have added in Table 6 the latency of different scheduling policies for RCS. Compared to a distance-based scheduling, our RCS introduces only negligible overhead while achieving significantly better performance. We can also see that accumulating prediction errors across all modules yields similar performance to our approach, but incurs additional computational overhead.
>
>
>
>
>
> ---------------
> > **Q2 : Is there a clear trade-off pattern for the method?**
>
> - We appreciate the insightful comment. We provide the trade-off of our method w.r.t. the threshold $\tau$ in Fig. 10 of Sec. D. By increasing $\tau$, RFC becomes more efficient, but with slightly degraded image quality.

---

### Author Response · Authors · 2025-11-20
**Global Response**

Dear AC and reviewers,

We sincerely appreciate your valuable time and effort spent reviewing our manuscript.

In this work, we propose Relational Feature Caching (RFC), a novel framework designed to accelerate diffusion transformers by exploiting the strong correlation between input and output features. As highlighted by the reviewers, our method is well-motivated (gkp8, h2W6, xGHL), insightful (gkp8, 6daa), novel (gkp8, xGHL), and is supported by comprehensive experiments and strong empirical results across diverse tasks (all).

We appreciate your constructive comments and suggestions. We have tried our best to address all concerns and carefully revised the manuscript. For ease of review, the modifications are temporarily highlighted in blue in the revised manuscript. The main revisions and enhancements can be summarized as follows:
- Improved presentation (Sec. 2, Sec. 3.1, Sec. 3.2, and Sec. 4.3).
- Extended empirical analysis with HunyuanVideo on VBench (Sec. B).
- Results on extreme acceleration ratios (Table 7 in Sec. C).
- Results on a distilled model (Table 8 in Sec. C).
- More details of RCS (Fig. 10, 12 in Sec. D, and Table 13 in Sec. E).
- Discussion on the CLIP score (Fig. 11 in Sec. D).
- Layer-wise analysis (Table 10 in Sec. D).
- Results on U-Net architecture (Table 11 in Sec. D).

We hope the revised version meets your expectations and look forward to your continued feedback.

Sincerely,

Authors.

---

### Meta-Review · Area_Chair_ScJm · 2026-01-04

**Summary:**

A way to speed up diffusion transformers is to cache features, especially ones that are computationally expensive, and reuse them at future timesteps in the diffusion process though extrapolation. This speeds up computation but introduces errors. The authors propose a method that exploits correlations between input and output errors to enhance the quality of predictions; the former is modeled as the difference of the input from its Taylor expansion, an approach previously used in TaylorSeer (Liu et al. 2025).

Reviewers appreciated both the theoretical analysis in arriving at the scheme, as well as the comparison to competitors. They have raised the following important concerns.


1) More experiments demonstrating where the method fails (input/output correlation is not as strong) were requested (gkp8,h2W6).

2) In a related vein, a question was raised on whether this correlation generalizes across architectures (6daa), including U-Net Based diffusion (h2W6) or distilled models (xGHL), and datasets (6daa,xGHL)

3) Several reviewers (6daa, xGHL,gkp8). asked about how $\tau$ is to be set; this controls a tradeoff between computation and accuracy.

4) Some reviewers raised issues of clarity (gkp8,6daa,xGHL)

5) A minor concerns about computational overhead was also brought up (xGHL).

6) W.r.t. CLIP score, TaylorSeer often performs better (h2W6)

Overall, I believe the authors have successfully addressed all these concerns in their rebuttal, as I explain below. I can understand why reviewers are lukewarm, especially the generalization power concerns 1+2.  I believe with the new experiments included the authors show that there is sufficient robustness to the approach, enough to warrant reporting this to the broader community.

**Reviewer Concerns:**

1) Reviewers have added more experiments demonstrating where the correlation is most challenged, including high acceleration scenarios. The method degrades but gracefully, and still outperforms competitors.

2) Multiple additional experiments with different architectures and datasets were included.

3) The ability to select $\tau$ is now being explored thoroughly.

4)-6) These are minor concerns, and the paper sufficiently addresses them.

**Reviewer Scores:**

One reviewer (xGHL) indicated that their comments were addressed, but they would keep their current scores. As everyone is leaning positive, I expect they would stay were they were or lean upwards.

---

### Decision · Program_Chairs · 2026-01-26

Accept (Poster)